# Multi-group analysis of grooming network position in a highly social primate

**Jonas R. R. Torfs**[1,2]*, **Jeroen M. G. Stevens**[1,2,3], **Jonas Verspeek**[1,2], **Daan W. Laméris**[1,2], **Jean-Pascal Guéry**[4], **Marcel Eens**[1], **Nicky Staes**[1,2]

**1** Behavioural Ecology and Ecophysiology Group, Department of Biology, University of Antwerp, Antwerp, Belgium, **2** Centre for Research and Conservation, Royal Zoological Society of Antwerp, Antwerp, Belgium, **3** SALTO Agro- and Biotechnology, Odisee University College, Sint-Niklaas, Belgium, **4** Zoological Park La Vallée des Singes, Romagne, France

\* jonas.torfs@uantwerpen.be

## Abstract

Individual variation in complex social behavioral traits, like primate grooming, can be influenced by the characteristics of the individual and those of its social group. To better grasp this complexity, social network analysis can be used to quantify direct and indirect grooming relationships. However, multi-group social network studies remain rare, despite their importance to disentangle individual from group-level trait effects on grooming strategies. We applied social network analysis to grooming data of 22 groups of zoo-housed bonobos and investigated the impact of three individual (sex, age, and rearing-history) and two group-level traits (group size and sex ratio) on five social network measures (out-strength, in-strength, disparity, affinity, and eigenvector centrality). Our results showed age-effects on all investigated measures: for females, all measures except for affinity showed quadratic relationships with age, while in males, the effects of age were more variable depending on the network measure. Bonobos with atypical rearing histories showed lower out-strength and eigenvector centrality, while in-strength was only impacted by rearing history in males. Group size showed a negative association with disparity and eigenvector centrality, while sex ratio did not influence any of the investigated measures. Standardization for group size did not impact the effects of sex and age, indicating the robustness of these findings. Our study provides comprehensive insights into the complexity of grooming behavior in zoo-housed bonobos, and underlines the importance of multi-group analyses for the generalizability of social network analysis results for species as a whole.

## Introduction

Many animal species live in social groups, which can be characterized by repeated interactions between group members that typically show a non-random pattern with more or less efforts directed towards specific partners [1]. Animals are thus embedded within a complex social network, consisting of all dyadic relationships among all group members [2–5]. As such, an individual's social relationships will not only depend on its direct connections (i.e. the number and strength of connections), but also on its indirect connections in the network (i.e. the level

**Funding:** J.R.R.T. (grant n° 1124921N), D.W.L (grant n° 11G3220N) and N.S. (grant n° 12Q5419N) were funded by Research Foundation Flanders (FWO, https://www.fwo.be/). The Flemish government provides structural support for the Centre for Research and Conservation of the Royal Zoological Society of Antwerp. The funders had no role in study design, data collection and analysis, decision to publish, or preparation of the manuscript.

**Competing interests:** The authors have declared that no competing interests exist.

of connectedness of their own connections) [6]. Social network analysis (SNA) provides a useful toolkit to study social systems and an individual's social position within the group, as it offers an opportunity to quantify various measures that represent different aspects of these direct and indirect relationships [2, 7, 8]. Several studies have shown the biological relevance of an individual's social network position based on both direct and indirect bonds, as it is directly linked to life-history traits and ecological processes such as longevity [9], reproductive success [9], and disease transmission [10]. For this reason, research efforts are now directed towards identifying which individual characteristics drive variation in individual social network measures [11], as ultimately, such studies can provide answers to more fundamental questions about the evolution of animal societies [12].

In primates, the time spent grooming others is often used as a proxy for the strength of their social bonds [13, 14]. Hence, SNA has been used to investigate individual variation in grooming behavior. An individual's direct connections in the grooming network can, for instance, be quantified through the strength measure, representing an individual's total grooming activity. Grooming strength can be further subdivided in its directional measures, the out-strength and the in-strength, which differentiate between outgoing and ingoing grooming interactions. Another direct measure is the disparity measure, which evaluates how grooming is divided among grooming partners, assessing the heterogeneity of an individual's grooming relationships. Besides direct social network measures, SNA also provides various measures of indirect connectiveness. These measures take into account the connections of your connections, representing a higher level of complexity regarding an individual's social behavior. For instance, affinity evaluates how strongly your connections are connected themselves, quantifying a preference for 'sociable' and 'popular' individuals in the network, while eigenvector centrality represents an individual's embeddedness within the entire grooming network as it is a measure for how well an individual is associated, as well as how well their connections are associated themselves (See Table 1 for an overview of the social network measures investigated in this study).

Both these direct and indirect grooming network measures are dependent on individual characteristics like sex [12, 17, 18], age [12, 19–21], and early social rearing conditions [22–24]. However, studies do not always yield consistent results for these associations, even when done on the same species. These inconsistent results might be attributable to the single-group nature of these studies. Grooming patterns in primates are highly variable and dependent on characteristics of the social group like size, sex ratio, stability, the number of infants present, and even social culture [13, 14, 18, 22, 25–28]. For instance, a meta-analysis of grooming rates across different primate species found that grooming frequency increased with group size as more social relationships need to be maintained, and decreased in groups with a female-biased

**Table 1. Overview of the social network measures investigated in this study.**

| Measure | Meaning in grooming network |
|---|---|
| *Direct measures of connectivity* | |
| Out-strength | Total frequency of grooming given |
| In-strength | Total frequency of grooming received |
| Disparity | A measure for the heterogeneity of grooming relationships, whether an individual's grooming relationships are equally strong or whether there are pronounced differences among the grooming bonds with different grooming partners [7]. |
| *Indirect measures of connectivity* | |
| Affinity | Tendency or preference of an individual to groom with high-strength individuals [15, 16]. |
| Eigenvector centrality | A measure of an individual's connectedness, as well as the connectedness of its associates, as a measure of social integration in the grooming network [11, 16]. |

sex-ratio [14]. In larger groups, individuals might also increase disparity as, due to time constraints, they will focus their grooming on specific partners instead of dividing it more equally among all group members [27]. Also, indirect measures such as affinity and eigenvector centrality can depend on group size. For example, in vervet monkeys (*Chlorocebus aethiops*), larger groups sizes led to lower individual eigenvector centralities in the proximity network [18], since larger groups tend to be more modular and less dense [28], making it more difficult for individual group members to obtain a central position in the network as a whole. Results from single group studies should thus not be generalized and considered representative for the species as a whole [29], and require validation in other groups belonging to the same species. Including multiple groups with varying group characteristics not only benefits representability of social network studies [30], but also increases statistical power to detect significant effects of individual characteristics like sex, age or rearing history on grooming network measures. Moreover, it allows us to disentangle the effects of individual characteristics on individual grooming measures from effects caused purely by group demographic differences like group size.

The aim of this study is to combine direct (out-strength, in-strength, and disparity) and indirect (affinity and eigenvector centrality) grooming network measures from a large multi-group ($N_{groups}$ = 22) sample of zoo-housed bonobos (*Pan paniscus*) to investigate to what extent factors at the individual (sex, age, and rearing history) and group-level (group size, sex ratio) explain variation in grooming strategies. Bonobos offer an interesting species for studying grooming network patterns as they live in large, multi-male multi-female social groups where grooming exchanges are used for the formation and maintenance of social bonds [31–36]. Bonobos show female dispersal (i.e., females disperse to another group once they reach sexual maturity) [37, 38] and non-exclusive female co-dominance, meaning that the most dominant individuals in bonobo societies are female, but not all females dominate all males [29, 39]. Female bonobos are able to dominate the physically stronger males by forming coalitions with other, usually unrelated female group members, which are maintained by grooming relationships [31], leading to high grooming frequencies within female dyads [40–42]. Males stay in their natal groups and their rank usually depends on the support they receive from their mother [37, 43, 44], and males form strong grooming bonds with their mothers, but also with other, often unrelated females in the group [40–43, 45]. Grooming within male dyads is less frequent than within female-female and male-female dyads [40–42].

Several non-SNA studies have investigated direct grooming measures in bonobos such as grooming frequencies (cf. strength) or grooming diversity (cf. disparity). Female bonobos tend to receive more grooming than males (cf. in-strength) [31, 46], likely because bonobos tend to groom high-ranking individuals [31, 47] and females tend to outrank most males [39]. Both sexes show similar grooming efforts towards others (cf. out-strength) and show similar grooming distribution patterns (cf. disparity) towards other group members [46]. Bonobos also show variation in grooming activities as they age, with higher levels of participation in grooming activities typically found in adults compared to subadults [47]. In old age, grooming activity is expected to lower again for two reasons. First, females with sexual swellings that signal an active reproductive status offer more popular grooming partners to both males and females [48], thus grooming activity is expected to be highest during the time-frame of reproduction, which in female bonobos lies roughly between 6 and 40 years old [49]. Second, as grooming tends to be directed towards higher-ranking individuals [31, 47], grooming interactions are expected to peak as individuals age and gain rank, but also lower again at old age when rank declines as individuals can no longer physically outcompete younger individuals. Moreover, age also affects the grooming distribution of bonobos: for example, young bonobos tend to be more selective in their grooming relationships as they focus more on kin [50] and adult

**Table 2. Predictions about the influence of individual and group level characteristics on the five social network measures that will be investigated in this study.**

| Measure | Predictions | |
|---|---|---|
| | Individual characteristics | Group characteristics |
| **Out-strength** | ∩-shape with age, lower in atypically-reared individuals | Increase with group size |
| **In-strength** | ∩-shape with age, lower in males and atypically-reared individuals | Decrease in females only in female-biased groups |
| **Disparity** | U-shape with age, higher in atypically-reared individuals | Increase with group size, decrease in female-biased groups |
| **Affinity** | U-shape with age, higher in males and atypically-reared individuals | Decrease in female-biased groups |
| **Eigenvector centrality** | ∩-shape with age, lower in atypically-reared individuals | Decrease with group size |

bonobos might become more selective in old age, in line with findings in chimpanzees (*Pan troglodytes*) [51]. However, changes with age in grooming behavior within adult bonobos remain to be tested as previous studies on bonobo grooming behavior did not include many elderly and post-reproductive (40+ years old) individuals. Very few studies have investigated indirect grooming relationships in bonobos using SNA, and all were based on single-group samples, limiting the generalizability of the results for the bonobo as a species. In a group of wild bonobos, older individuals were more likely to initiate and lead group movements, as were individuals with higher eigenvector centralities [52], suggesting a potential correlational pattern between age and eigenvector centrality that remains to be tested. Furthermore, in a zoo-housed group, bonobo males and females did not differ in strength or eigenvector centrality, but wild-caught individuals showed higher strength and eigenvector centrality than mother-reared, captive-born bonobos, indicating an effect of rearing history on social network position in bonobos [53].

Based on this, we predict the following effects of individual and group-level traits on individual grooming network measures (see Table 2 for a summary). We expect female bonobos to have higher in-strength than males, but we expect no sex differences in out-strength, disparity, or eigenvector centrality. We predict males will have higher affinity compared to females, because males tend to prefer females as grooming partners [40–43, 45], and usually occupy the lowest ranks in the hierarchy and might therefore focus their grooming on high-ranking individuals that are expected to have high strengths [31, 47]. Furthermore, we expect age-effects on out-strength, in-strength, and eigenvector centrality to follow a reverse-U shape in both sexes, with an increase in these measures from subadult to adult, with a maximum around their prime age (± 25 years old), and a decrease in elder bonobos. For disparity and affinity, we expect a U-shaped relation to age with higher values in young and old bonobos. Finally, for rearing history, we expect that atypical rearing histories can cause a deficit in the development of social skills, leading to atypically-reared individuals showing longer strength, lower eigenvector centrality, and potentially higher disparity and affinity compared to mother-reared individuals, as found in chimpanzees [22, 23]. As our study includes a large age-range of atypically-reared individuals, we should also be able to disentangle the potential influence of age and rearing history on grooming strategies that previous single-group studies could not do [53].

Considering group characteristics, we predict that individuals living in larger groups will show higher out-strength as the number of grooming interactions increases when more social relationships need to be maintained [14]. Group size is not expected to impact in-strength as the amount of grooming received will rather depend on individual characteristics such as sex, age, and rearing history. In larger groups, disparity might increase, as individuals will have less time available to groom all groups members and might thus focus their grooming on certain individuals (e.g., high-ranking individuals [31]). Affinity is expected to be independent of group size as the preference of individuals to associate with high-strength partners is likely

more dependent on the rank and reproductive status of the receiver rather than group size. Eigenvector centrality is expected to decrease with group size, as larger groups will lead to more modular social networks [28]. For sex ratio, we expect female in-strength to lower as groups become more female biased as females overall receive more grooming than males. Out-strength is expected to be unrelated sex ratio, because both sexes show similar frequencies of grooming given [46]. We expect disparity to be lower in more female biased groups as both sexes prefer female partners to groom with. We expect no influence of group sex ratio on eigenvector centrality in bonobos, since previously no sex differences were found in eigenvector centrality [53]. we would expect individuals in more male-biased groups to show overall higher affinity rates, following our predictions for sex-specific effects on affinity.

## Materials and methods

### Study subjects and housing

Behavioral data were collected between November 2011 and July 2022 in 8 different zoological institutes across Europe. All bonobos were housed adherent to the guidelines of the EAZA Ex situ Program (EEP). Some institutions were visited multiple times during this period, and some institutions housed multiple groups, such that in total 22 compositionally different groups of bonobos were observed. A group was considered to be compositionally different when they differed in the presence or absence of at least one individual [22, 54] (S1 File), since previous research has shown that the removal from or addition to the network of one or more individuals can lead to changes in network structure and the position of some or even all group members [18, 54–57]. All groups consisted of individuals of both sexes, and some groups also contained juveniles and/or infants (i.e., individuals under 7 years old). Juveniles and infants were not included in the networks as we did not collect reliable data on their grooming behavior. Group sizes varied between 3 and 15 (median = 6) individuals over 7 years old (hereafter referred to as adults), which is comparable to previous SNA studies on zoo-housed great apes, where group sizes typically ranged from 3 to 14 individuals [15, 22, 23, 58, 59]. Adult male-to-female sex ratios varied between 0.20 and 2.00, with 14 groups being female-biased (i.e., sex ratio below 1), 3 groups being male-biased, and the remaining 4 groups having an equal number of adult males and females (S1 File and S1 Table in S2 File). Because some individuals were transferred between groups over time, and in some institutions, groups were observed multiple times over the 11 year time-span, certain individuals were observed more than once. In total, 84 different individuals were observed, of which 48 were observed once, 25 were observed twice, 6 were observed 3 times, and 5 were observed 4 times, each time in different group compositions (S1 File). This resulted in a total sample size of 136 datapoints. Behavioral data were collected by 11 different observers using a standardized ethogram, who were subjected to rigorous training for at least 2 weeks and tested for inter-observer reliability by scoring the same two 10 minute bonobo behavioral videos and reached a mean $r = 0.85$ across all observers, indicating high reliability of observations [60]. Information on the sex, age, and rearing history of individual bonobos was obtained from the EEP Studbook [61]. Regarding rearing history, we differentiate between atypically-reared and mother-reared bonobos. Atypically-reared bonobos include all bonobos that were separated from their mother and possibly other conspecifics at an early age, including wild-caught and nursery-reared individuals. A summary of the study sample can be found in Table 3.

### Behavioral data collection

Grooming data were collected using group scan sampling [62] with an identical protocol in all groups, such that all networks were directly comparable [11]. Using a scan sampling method

**Table 3. Sample size within each sex and rearing history category, and median age and age range for each sex.** Number of unique individuals within each category is denoted between brackets.

|  | Total | Mother-reared | Atypically-reared | Mean age (years) | Age range (years) |
|---|---|---|---|---|---|
| **Males** | 51 (33) | 39 (24) | 12 (9) | 19 | 7–45 |
| **Females** | 85 (51) | 62 (39) | 23 (12) | 24 | 7–71 |

to collect data for our social networks was deemed appropriate, since our zoo-housed subjects had stable group compositions within observation periods, small group sizes, and were easily visible at all times [63], which also reduced sampling bias in our dataset to a minimum [30]. A scan was done approximately every 10–15 minutes, resulting in a mean of 626 scans per group (range 274–1103). During a scan, the behavior that each group member was performing at that moment was recorded using a standardized ethogram on a laptop with the Observer XT software (Noldus, the Netherlands). Data on agonistic interactions were recorded ad libitum in-between group scans and were subsequently used for rank analysis (see below). From the group scans, we extracted all grooming data. A grooming interaction was defined as each instance where a subject manipulates the receiver's body surface and hair with its fingers or lips. Mutual grooming events were inserted as two separate interactions (one from A to B and one from B to A). On average, 316 grooming interactions were observed per group (range 116–728), indicating that enough data had been collected in each group to construct reliable social networks [64].

Based on these grooming data, we constructed a directed, weighted grooming matrix for each group, meaning that each group represents their own separate social network. Since the total observation time differed among groups, the grooming matrix was standardized [11] using the 'index of interactions' approach [65]: the number of scans where individual A groomed individual B was divided by the total number of scans done in their group, excluding the numbers of scans where individual A and B were simultaneously out of sight. When both individuals were invisible to the observer, it was unknown whether they were grooming or not, and therefore these instances were excluded when constructing the grooming matrices.

### Statistical analysis

**Social network analysis.** From these grooming matrices, we calculated five network measures for all individuals within their respective network: the out-strength, in-strength, disparity, affinity, and eigenvector centrality (Table 1). All measures were calculated from the weighted networks. The out-strength, in-strength, and affinity were calculated using the ANTs package [66] in the R environment (www.r-project.org, version 4.2.2). Disparity was calculated manually following the formula used in Kalcher-Sommersguter et al. [23]. Eigenvector centrality was calculated using the igraph package [67] in R.

Prior to further analysis, we tested to what extent the grooming network measures correlated with each other. Correlations among different network measures could be high as these are all calculated from the same object (i.e., the grooming matrix), which excludes the use of different network measures in the same model and can influence the interpretation of the results of different models using the social network measure as a response variable [68]. Following the suggestions from Webber et al. [68], we a priori decided to exclude network measures if they showed a correlation coefficient with an absolute value above 0.70 with other network measures. Correlations among all five grooming network measures were assessed using Pearson's rank correlations in R. We found that all network measures were correlated significantly with at least one other network measure (S2 Table in S2 File), with the strongest

correlation found between out-strength and affinity (r = -0.63, p < 0.001). We decided to investigate all five social network measures further in this study for three reasons: (1) the absolute values of all correlation coefficients were lower than 0.70, (2) all network measures will be used in different models, avoiding violation of statistical assumptions, and (3) each network measure represents a separate aspect of grooming behavior, and therefore, studying the different measures represents a more complete overview of bonobo grooming behavior which will aid in interpreting the results. This also allows for comparison of SNA measures with traditional grooming variables published in literature for validation purposes (such as strength and disparity), while other SNA measures have not been investigated in so much detail in bonobos and thus allow for novel insights and contributions to current grooming literature (such as eigenvector centrality and affinity).

**Factors associated with individual grooming network measures.** We used linear mixed models (LMM's), implemented within the lmerTest package [69] in R, to study the influence of sex (male or female), age, rearing history (mother-reared or atypically-reared), group size (number of adult individuals), and sex ratio (proportion males to females) on the five social network measures. We constructed separate models for each social network measure. In each model we included sex, age, rearing history, group size, and sex ratio as fixed factors, as well as the two-way interactions between sex and the other variables, since it is possible that these effects differ in a sex-specific manner. Age was modelled in both a linear and quadratic fashion to investigate whether age-related social network measures follow a quadratic pattern. Age was centralized around the mean and then squared to avoid collinearity between age and its square in the models. Individual identity was added as a random intercept in all models to control for repeated measurements of the same individual, and group identity was added as a random intercept to control for the non-independence of the data within the same group [66]. For the direct SNA measures (out-strength, in-strength, and disparity), we ran models on the full dataset, while for the indirect SNA measures (affinity, eigenvector centrality), groups with less than 5 individuals were excluded as these indirect measures are not biologically meaningful in such small networks ($N_{indirect}$ = 126 individuals from 20 groups). We tested multicollinearity of the variables using the variance inflation factor (VIF), with the threshold placed at VIF > 5 [70], and no multicollinearity was detected. Outliers were detected using the Rosner's test in the EnvStats package [71], which led to the removal of one outlier in the in-strength dataset and two outliers in the affinity dataset. Shapiro-Wilk tests and diagnostic plots (residuals vs. fitted values and QQ plots) showed no violations of the assumptions of normality and homogeneity of variances. We first ran full models and then reduced the models based on the Akaike Information Criterium (AIC), such that the most parsimonious model with the lowest AIC value remained [72]. Model summaries were used to determine the parametric coefficients of the fixed effects in the remaining models. We refrained from using node-based permutation tests to test significance of the variables as these have been reported to not perform better than parametric regression in terms of returning false positives and negatives [73]. In the case of multiple comparisons (e.g., when investigating a significant rearing-by-sex interaction), post-hoc comparisons were done with Tukey-adjustments using the emmeans package [74].

If an effect of group size was found on a certain network measure, we re-ran the model of this network measure after standardizing it for group size, to test whether any other associations still remained and whether they were independent of the effect of group size. To standardize in-strength and out-strength, we divided the measure by the number of available grooming partners (i.e. N-1, with N being the group size) [7]. Disparity was standardized by calculating the 'deviation from edge weight disparity', following the methodology of Kalcher-Sommersguter et al. [23]. Eigenvector centrality and affinity were standardized by scaling the value of each individual to the maximum value within their network [54]. Thus, the maximum

in each network has a value of 1 and all other values are scaled towards this value. Subsequent analyses were performed as described above with separate models for each network measure, except that group size was removed as a predictor from all models.

**Rank analysis.** To visually assess how dominance rank changed with age in each sex, as this could be linked to age-effects on grooming behavior, we constructed dominance hierarchies for those groups for which sufficient and reliable data was available. We used normalized David's scores (DS) as a measure for individual dominance rank [75]. For each group, a flee-upon-aggression matrix was constructed, using the number of times an individual won an agonistic interaction from another individual (i.e., when the opponent fled from the attacker) as behavioral indicator for agonistic dominance [32]. For each group, we used randomization tests to assess whether the dominance hierarchy showed significant steepness and linearity. Individual David scores, steepness and the significance of the steepness of the dominance hierarchy was assessed using the steepness package [75] in R. Linearity was measured as the Landau's linearity index (h'), which corrects for unknown or tied relationships [76]. h' was calculated and its significance was assessed using the DomiCalc software [77]. For both measures, α was set at 0.10 to enlarge our dataset. These software programs require data on aggressive interactions between all dyads in the group to produce significant results. However, in reality, submissive behavior and aggression were rare in our groups and definitely less frequent and even sometimes absent in female-female dyads. With higher observational zeroes (e.g., dyads with no interactions) in the flee-upon-aggression matrix, it is difficult to obtain a significant linear dominance hierarchy because some dyads will occupy similar rank positions. However, if we were to apply a more stringent approach (e.g., alpha set at 0.05), we would have only two groups left which does not allow for a representation of sex-specific rank correlations with age. With this limitation in mind we decided not to include our rank results into our SNA models, but rather use them as a visual aid to help interpret the age-related changes in grooming. Five groups passed our more relaxed criterion (S1 Table in S2 File), creating a dataset of 40 individuals to visually assess the correlation between age and agonistic dominance rank based on DS. DS were mean-centered within each group to account for differences in group size [78].

## Ethics statement

As this is an observational study, the scientific advisory board of the Royal Zoological Society of Antwerp waivered the need for ethical approval. This study conformed to the ASAB guidelines for animal behavior research.

## Results

### Effect of individual characteristics

Age had an influence on all grooming network measures (Fig 1). In-strength and disparity changed with age in a quadratic fashion (In-strength: β = -0.00004 ± 0.00001 SE, t(118.3) = -2.432, p = 0.016; Disparity: β = 0.00012 ± 0.00004 SE, t(74.7) = 3.108, p = 0.003). In-strength was lowest in young and old bonobos, while disparity showed a reverse pattern, being higher in younger and old bonobos. For out-strength, affinity, and eigenvector centrality, the effect of age differed between the sexes, as indicated by significant age-by-sex interactions. In females, a quadratic relationship with age was found for out-strength (β = -0.00005 ± 0.00002 SE, t(123.0) = -3.005, p = 0.003) and eigenvector centrality (β = -0.00014 ± 0.00006 SE, t(63.7) = -2.393, p = 0.020), with highest values for both measures found in middle-aged females. In males, out-strength declined linearly with age (Out-strength: β = -0.002 ± 0.001 SE, t(128.3) = -2.911, p = 0.004), indicating that for each 10-year increase in age, males decrease their time

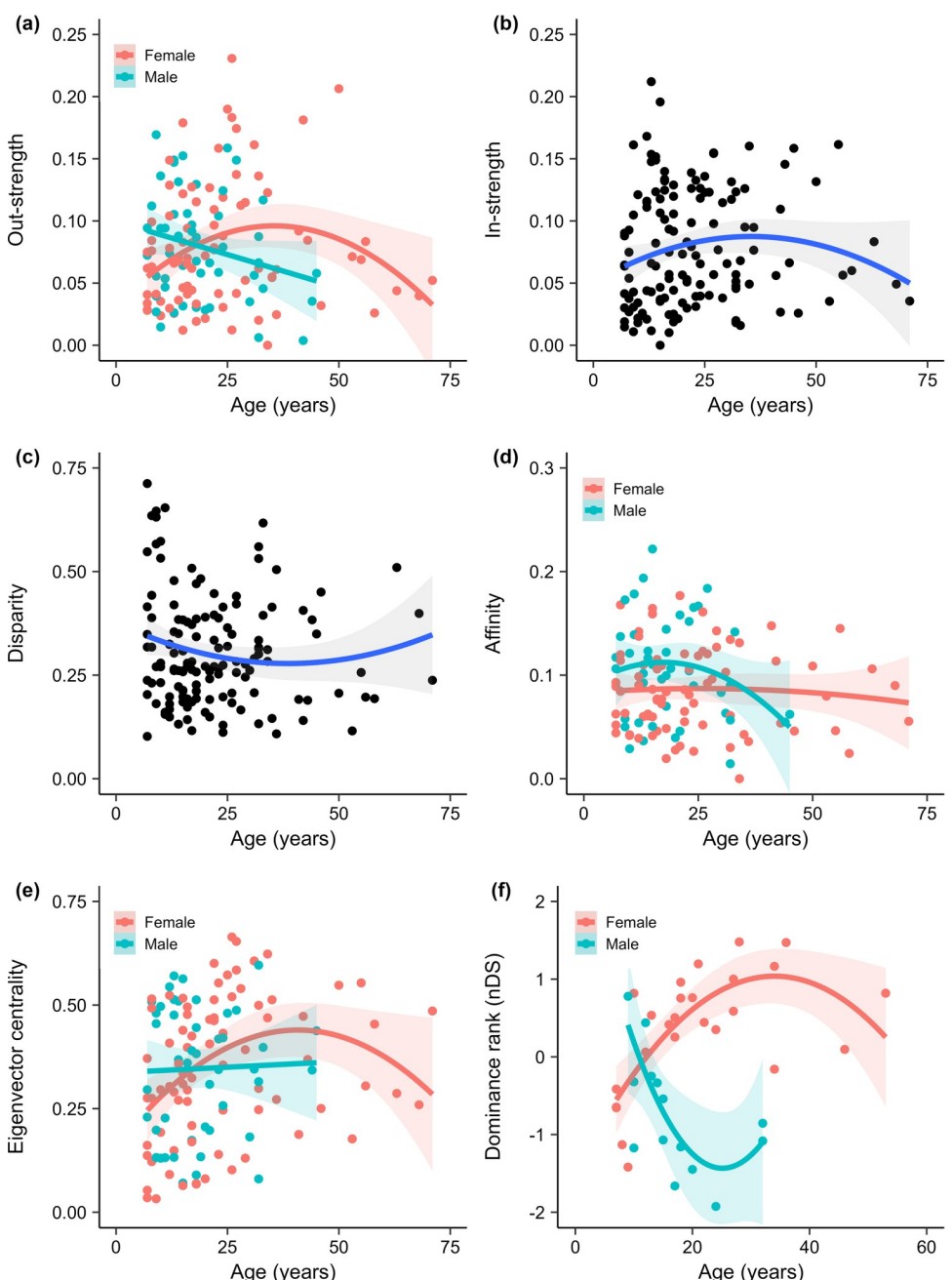

**Fig 1. The effect of age on the different grooming network measures.** (a) Out-strength, (b) in-strength, (c) disparity, (d) affinity, (e) eigenvector centrality, and on (f) agonistic dominance rank as measured by normalized David's scores (nDS). Black dots represent individual datapoints. If an age-by-sex interaction was found, we indicated males in blue datapoints with a blue trendline, while females are indicated in red. Shaded area represents the 95% confidence interval.

spent on grooming others with 2%. To put this in perspective, the average out-strength for males across all ages is 0.079, indicating that about 8% of their time is spent on grooming others. Eigenvector centrality slightly increased with age in male bonobos, although this slope was not significant ($\beta = 0.002 \pm 0.003$ SE, $t(78.7) = -1.627$, $p = 0.108$). Affinity showed a tendency towards a quadratic relationship with age in male bonobos, with the rate of decline being

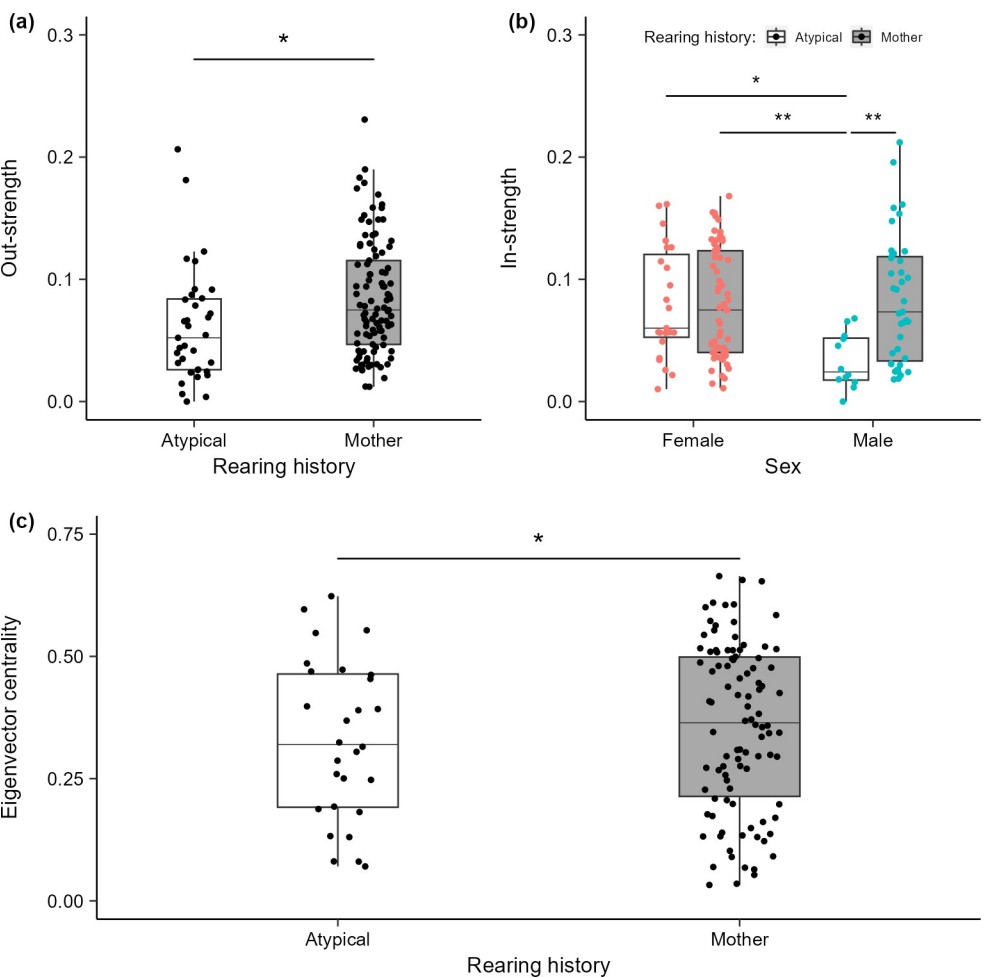

**Fig 2. The effect of rearing history on grooming network measures.** Rearing-history significantly affected (a) out-strength, (b) in-strength, and (c) eigenvector centrality. Boxplot figure with lower and upper box boundaries at 25th and 75th percentiles, respectively. Line inside box shows median, black dots show individual data points for both sexes, while red dots represent females and blue dots represent males in case of rearing-by-sex interactions. * P < 0.05, ** P < 0.01, *** P < 0.001.

shallow in younger bonobos, but accelerating with age, with the lowest values found in old male bonobos (β = -0.00010 ± 0.00005 SE, t(62.2) = -1.917, p = 0.060). Affinity show a weak non-significant quadratic relationship with age in female bonobos (β = -0.000009 ± 0.000015 SE, t(57.6) = 0.583, p = 0.562).

Rearing history influenced out-strength, in-strength, and eigenvector centrality (Fig 2). Mother-reared bonobos had higher out-strength and higher eigenvector centrality than atypically-reared individuals (Out-strength: β = 0.027 ± 0.011 SE, t(129.6) = 2.488, p = 0.014; Eigenvector centrality: β = 0.102 ± 0.039 SE, t(88.8) = 2.636, p = 0.010). For in-strength, the effect of rearing history showed an interaction with sex (t(116.1) = 2.613, p = 0.010; Fig 2B): atypically-reared males had lower in-strength than mother-reared males (β = -0.051 ± 0.015 SE, t(127) = -3.410, p = 0.005), mother-reared females (β = -0.050 ± 0.014 SE, t(128) = -3.461, p = 0.004) and atypically-reared females (β = -0.046 ± 0.016 SE, t(116) = 2.854, p = 0.026). There was no difference between all other sex-rearing categories (all p > 0.05; S3 Table in S2 File). Tables with test statistics for the fixed effects and interactions for the best-fitting model per network measure can be found in S4 Table in S2 File.

Since all social network measures showed a relationship with age, we additionally assessed visually whether agonistic dominance rank showed age-related changes (Fig 1F). In females, a quadratic relationship was found, with highest dominance rank reached around 35 years old and lower dominance ranks in younger and older females. In males, a U-shape could be observed, with highest dominance rank seen in the youngest males in our dataset, with a minimum around 25 years old. In older males, dominance increases again, but this effect might be driven by two older males from the same group (see S1 File).

## Effect of group characteristics

Group size significantly influenced disparity and eigenvector centrality. Disparity showed a negative relationship with group size ($\beta$ = -0.025 ± 0.008 SE, t(17.1) = -3.120, p = 0.006; Fig 3A). For eigenvector centrality, a negative effect of group size was found for both sexes, but in females ($\beta$ = -0.030 ± 0.005 SE, t(114.8) = -5.798, p < 0.001) the effect was stronger than in males ($\beta$ = -0.007 ± 0.001 SE, t(112.5) = 2.608, p = 0.010; Fig 3B).

Since disparity and eigenvector centrality were significantly influenced by group size in both sexes, we set out to test whether the influence of age and rearing history on these measures still held after standardizing these measures for group size. This led to the same results (see S5 Table and S1 Fig in S2 File).

## Discussion

The aim of this study was to investigate to what extent individual-level and group-level factors predict individual variation in bonobo grooming behavior, using SNA on a large multi-group dataset. The results show sex-specific age effects on all social network measures. In female bonobos, all grooming network measures followed a quadratic relationship with age, except for affinity, which showed no clear relationship with age. For males, the effect of age was more variable among different measures. Rearing history affected the strength-measures and eigenvector centrality: out-strength and eigenvector centrality were lower in atypically-reared bonobos, while in-strength was lower in atypically-reared males compared to females and mother-reared males. The effects of group-level variables on grooming network measures were less strong: sex ratio did not influence any of the network measures, while group size influenced

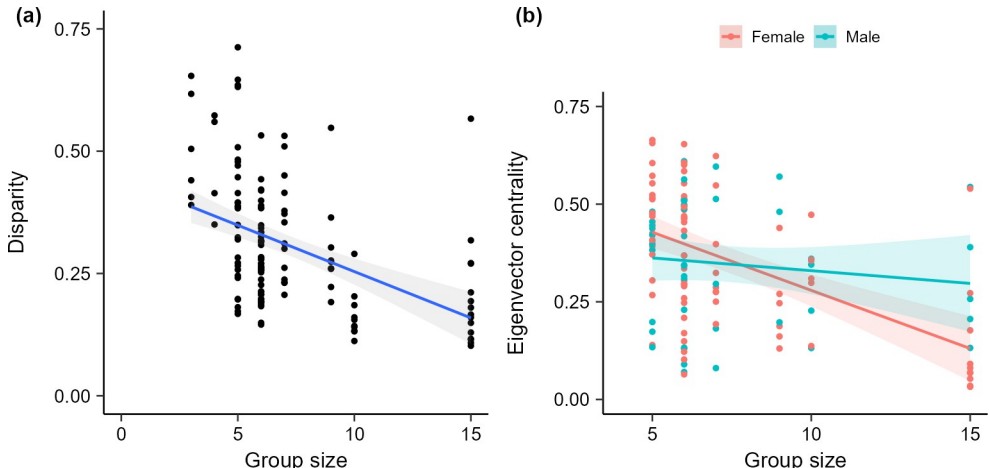

**Fig 3. The effect of group size on social network measures.** Group size significantly affected (a) disparity and (b) eigenvector centrality. Black dots show individual data points for both sexes, while red dots represent females and blue dots represent males in case of group size-by-sex interactions. Shaded area represents the 95% confidence interval.

disparity and eigenvector centrality. After correcting for group size, individual-level effects on disparity and eigenvector centrality remained identical, indicating that individual-level traits have robust effects on determining grooming strategies.

## Effects of individual characteristics

Following our predictions, in-strength showed a reverse U-shape with age, with lower levels of grooming received in young and old bonobos, with a peak in grooming frequencies received seen around age 30. Out-strength showed the same pattern in female bonobos. For females, this age-effect is likely partly attributable to status-effects, as our visual assessment revealed that dominance rank follows a similar pattern with age in female bonobos. Higher-ranking bonobos often are popular grooming partners [31, 47, 79], a pattern reflected here in both female strength measures. In addition, females with sexual swellings are more attractive grooming partners for both sexes [48], likely resulting in higher levels of female grooming interactions during the fertile period in life, which begins at the age of first menarche between age 6 and 11.25 years [80, 81], and ends when genital swelling cycles cease, which falls approximately between the ages of 40 and 50 years in zoos [49].

The effect of age on in-strength in male bonobos is less likely to depend on status-effects, as dominance rank shows a U-shape with age for males, opposite to the pattern found for in-strength. Thus, in males, other factors besides rank are likely responsible for these changes in in-strength with age. The fact that in-strength declines for older males might be a consequence of the fact that male bonobos receive most grooming from their mothers [34, 41, 43]: as males age, chances are higher that their mother has died, which could explain lower in-strengths for older males. Moreover, out-strength decreased with age as well but in a linear fashion, meaning that males also groom others less often as they get older. In chimpanzees, a similar pattern is found in females which has been related to their subordinate status, as they try to avoid falling victim to agonistic interactions [21]. As male bonobos, and especially older males, are generally subordinate to their female group members (Fig 1F) [32, 39, 47], this could also explain their reduced levels of affiliation in our study.

We did find an impact of early social rearing history on out-strength, eigenvector centrality, and on male in-strength. Atypically-reared individuals, regardless of their sex, showed lower rates of grooming given compared to mother-reared bonobos, indicating reduced social interactions in these individuals. These individuals were also more peripheral in the grooming network and played a less important role in maintaining network cohesion, as indicated by lower eigenvector centrality values. These rearing-effects are opposite to what was previously reported for bonobos, where wild-caught individuals were found to have higher overall strength and eigenvector centrality than captive-born individuals [53]. However, a confound might have been present between age- and rearing-effects in this previous study, as all female wild-born bonobos were around 30 years old, which corresponds to the age of maximum eigenvector centrality in our study. Rearing conditions of wild-caught individuals also show large variability (for example, in the age at which individuals ended up in captivity, in the number of years they were hand-reared, in which conditions they were raised, etc.) that can have biasing effects in small single-group studies. In our multi-group sample, a large variety of backgrounds and ages was included in the atypically-reared category, thereby limiting potential sampling bias. Our results also replicate several findings from chimpanzee studies, where individuals that were atypically-reared showed reduced levels of social contacts. For example, wild-born chimpanzees showed lower out-strength, as did individuals that were mainly housed without conspecifics during infancy or maternally deprived during early life [22, 23]. For in-strength, an effect of rearing history was found only in males, with atypically-reared males

receiving less grooming than females and mother-reared males. This is likely attributable to the strong grooming bonds found in mother-son dyads [34, 41, 43], leading to lower rates of grooming received for motherless individuals. Interestingly, no difference was found between mother-reared and atypically-reared females in in-strength, indicating that early social rearing history does not impact the popular status of adult females as grooming partners for other group members. As females are the dispersing sex in bonobos [37, 45, 82, 83], adult females do not have the opportunity to maintain strong grooming bonds with their mothers. They rather develop grooming bonds with unrelated males, unrelated females, and eventually their own sons [34, 41, 43]. This process of female migration is mimicked in European zoos by transferring young females to a new group when they reach reproductive age [49].

Disparity showed a positive quadratic relationship with age, indicating that younger and older bonobos will be more selective in their grooming partners. Young bonobos often focus their grooming on kin [50], indicating higher selectivity at younger ages. However, young females that have recently been transferred into a group cannot focus their grooming on kin. These newly-arrived females often target their grooming towards older, higher-status females in the group [43, 50, 84], which can explain their higher disparity. With age, disparity declined, but in old age, disparity increased again, showing bonobos become more selective again after their thirties. These results are in line with a recent body of literature reporting similar effects of higher social selectivity with age across a variety of primate species [e.g., 19, 51, 85].

Affinity of male bonobos was overall higher than female affinity and declined as individuals aged, indicating that younger males have a higher tendency to groom high-strength individuals than older males. This indicates that males, more than females, tend to direct their grooming towards high-strength individuals, like their mothers, higher-ranking females, and/or cycling females [31, 47, 48, 79]. The decline throughout life, which increased in acceleration in old age, might be attributable to older males no longer having their mothers in the group or their mothers losing strength as they age. Eigenvector centrality showed different patterns in male and female bonobos. In females, eigenvector centrality varied with age in a negative quadratic fashion, similar to their grooming strength, with a peak found when females are high-ranking, cycling and/or have offspring in the group. The link between eigenvector centrality and social status has been confirmed in other primate species (e.g., [24, 86]), and likely explains the association found in wild bonobos where more central individuals initiated more group departures [52]. In males, eigenvector centrality did not change significantly with age. For male bonobos, other factors might determine eigenvector centrality, such as for example the eigenvector centrality or status of their mother with whom they are strongly connected in the network.

## Effects of group characteristics

The two group traits included in this study, group size and sex ratio, had limited effects on individual grooming metrics. In contrast to another study reporting an impact of sex ratio on grooming behavior [14], our study did not support this claim in bonobos. However, this result is to be treated with caution as the majority of groups in our study were female-biased (14/22 female-biased groups versus 3/22 being male-biased), and there was a confound between group size and sex ratio, with the largest groups in our study all being female-biased (S1 File and S1 Table in S2 File). This is important to consider when interpreting the results, as for some network measures, opposite effects between group size and sex-ratio would be expected in large male-biased groups (Table 2), although these are rare in bonobos due to the lower life-expectancy of male bonobos [49, 87]. Moreover, due to the relatively low power our study has on the group-level, a possible effect of sex ratio on grooming network position might have

gone undetected. Still, it needs to be noted that a previous study did not find any effect of group sex ratio on grooming patterns in wild bonobos [88], so perhaps bonobo grooming strategies are relatively independent of the sex ratio of the group.

Group size effects were found to influence two of the five SNA measures: disparity and eigenvector centrality. In contrast to our predictions, disparity decreased with group size, meaning that in larger groups, individuals became less restricted in their grooming relationships. A similar pattern was found in a meta-analysis of different primate groups [7]. This might be a consequence of the fact that out-strength is unrelated to group size, and our rather small networks having high densities (i.e., the ratio of the number of actual connections to all potential connections in the network [66]), meaning that almost all individuals have a grooming relationship with all other group members. If individuals might dedicate a certain amount of their time to grooming, and if they want to maintain a certain minimal grooming relationship with all members of their group, they will have to be less selective in larger groups compared to smaller groups in order to maintain these minimal grooming bonds, as their out-strength did not increase with group size. In smaller groups, individuals might have more grooming time available to become selective towards certain group members and to show larger selectivity in their grooming relationships by grooming their preferred grooming partners more often.

In line with our expectations, eigenvector centrality significantly decreased with group size. Larger groups have less dense and more modular networks [28], leading to lower eigenvector centralities for individual group members. However, it is interesting that the effect differed between the sexes, with a larger effect in females than in males. Females show a greater reduction in eigenvector centrality with group size than males do, which could be explained by differences in social patterns between the sexes. Females will form strong grooming bonds with all group members, while male bonobos are more selective in their grooming bonds and tend to prefer high-strength female group members to groom with. Hence, when group size increases and thus the absolute number of females in the group increases, females will more drastically drop in centrality as more equally central females are present, while males will have more high-strength females available to associate with, thereby showing a less drastic reduction in centrality. Moreover, the finding that group size effects on eigenvector centrality differ between males and females illustrates why multi-group studies are important. As seen in Fig 3B, in smaller groups, females tend to have higher eigenvector centrality than males, while in the larger groups, males have higher eigenvector centrality than females. If only one group is studied, a sex-difference in eigenvector centrality might be a consequence of the group size and not reflect an absolute difference in social embeddedness between the sexes.

While these findings illustrate that group size can impact individual-level social network measures, their effects could be eliminated altogether by standardizing our measures for group size. As the effects of our individual level variables (sex and age) on grooming metrics remained identical after correction for group size, they appear to have a robust effect on grooming behavior that is independent of demographic variation.

## Limitations and future directions

In our analysis, we considered groups that differed in the presence or absence of at least one group member as fully separate networks, as this could have considerable effects on network structure [22, 54] and thus their networks could be considered as independent. Still, different groups with a large proportion of the same group members might not be fully independent of each other, as tenure is known to influence social bonds in bonobos, with stronger social relationships found in individuals that have spent more time together [34, 53]. Thus, we might

expect that groups sharing certain group members might not be fully independent of each other. However, it is difficult to control for this in our models as data were modelled at the individual level. Future studies should investigate how variation in dyadic tenures among individual group members structure social networks as a whole, and what the effects are of introducing new animals or removing certain group members on the structure of the social network.

Another limitation of our study are the small sizes of some of the networks, which could influence indirect social network measures such as eigenvector centrality and affinity. In small networks, these measures are less informative as they usually show high densities with connections among most, if not all, dyads within the group. However, small networks are inherent to the study of SNA in zoo-housed great apes and our networks were similar to group sizes from previous studies that typically ranged from 3 to 14 individuals [15, 22, 23, 58, 59]. Therefore, by including networks ranging from 3 to 15 individuals (5 to 15 in the case of indirect measures) in our study, we span the variation seen in previous studies which allows for direct comparisons. Moreover, the broad range in group sizes allows for broad testing of group size effects on social network position.

As the study of animal social networks is a rapidly developing field, especially regarding advances in statistical methodology, future studies on primate grooming networks could aim to include more state-of-the-art analytical approaches to investigate variation in grooming network measures. For example, besides using parametric models or permutation methods, generative models can be used to study social network data as well, as they account for the non-independence of social network data [89]. In addition, as sampling effort varies across groups, but also within groups (e.g., certain dyads might be observed more often than others and hence their grooming relationship might be more 'certain'), uncertainty in network edges could be taken into account in downstream analyses. Instead of correcting for variation in sampling effort using commonly used social network indices, using other frameworks such as the newly developed BISoN framework [90] could provide a way to incorporate sampling effort and edge uncertainty into down-stream analyses. In our current study, sampling effort was relatively even across groups and across dyads within the groups due to the high visibility of individuals, reducing edge uncertainty. Nonetheless, future studies could take advantage of these newly developing analytical techniques to more reliably investigate primate social networks.

Ideally, our analyses should be repeated in wild bonobo groups, as certain aspects of the captive environment are known to influence grooming rates. For example, we did not account for density (i.e., number of individuals per surface area in the enclosure), even though higher densities could lead to increases in grooming rates [e.g., 91]. In the wild, group size will be independent of available space, and as such, grooming rates and network structure will not depend on density effects, allowing for more reliable investigation of the social implications of group size.

## Conclusions

In conclusion, our study shows the impact of both individual and group-level variables on both direct and indirect individual grooming network measures of bonobos, and thereby illustrates the primary role of traits related to social and reproductive status like sex, age, and rank in predicting individual grooming strategies. While rearing history influenced an individual's grooming given, it only affected the popularity of males as grooming partners and did not influence the popularity of females. Due to the inclusion of both direct and indirect measures of social behavior, our study provides novel insights into the complexity of bonobo grooming

behavior, and underlines the importance of multi-group analyses for the generalizability of SNA results for the species as a whole.

## Supporting information

**S1 File. Detailed data on group composition, individual metadata, and raw adjacency matrices.** For each individual, data is included on sex, age, and rearing history, as well as data on the five social network measures investigated in this study. For each group, data was included on the group size and sex ratio.
(XLSX)

**S2 File. Contains all supporting tables (S1-S4 Tables) and figures (S1 Fig).**
(DOCX)

## Acknowledgments

We thank the participating zoos and bonobo care staff for hosting us for behavioral data collection and for supporting this work: Apenheul (Apeldoorn, the Netherlands), La Vallée Des Singes (Romagne, France), Ouwehands Dierenpark (Rhenen, the Netherlands), Twycross Zoo (Twycross, United Kingdom), Wilhelma Zoological and Botanical Garden (Stuttgart, Germany), Wuppertal Zoo (Wuppertal, Germany), Zoo Frankfurt (Frankfurt-Am-Main, Germany) and Zoo Planckendael (Mechelen, Belgium). We also thank all the students that were involved in data collection: Ilke Fromont (University of Antwerp), Adriana Solis (University of Groningen), Sandra Castells (University of Gerona), Annemieke Podt, Sanne Roelofs, Wiebe Rinsma, Marloes Borger, Martina Wildenburg and Sjoerd Beaumont (all Utrecht University).

## Author Contributions

**Conceptualization:** Jonas R. R. Torfs, Nicky Staes.

**Data curation:** Jonas R. R. Torfs, Jeroen M. G. Stevens, Nicky Staes.

**Formal analysis:** Jonas R. R. Torfs.

**Funding acquisition:** Jonas R. R. Torfs.

**Investigation:** Jonas R. R. Torfs, Nicky Staes.

**Methodology:** Jonas R. R. Torfs, Jeroen M. G. Stevens, Nicky Staes.

**Resources:** Jean-Pascal Guéry.

**Supervision:** Marcel Eens, Nicky Staes.

**Visualization:** Jonas R. R. Torfs.

**Writing – original draft:** Jonas R. R. Torfs, Nicky Staes.

**Writing – review & editing:** Jonas R. R. Torfs, Jeroen M. G. Stevens, Jonas Verspeek, Daan W. Laméris, Marcel Eens, Nicky Staes.

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
