## [Decision Letter · Decision Letter 0]

12 Dec 2022

PONE-D-22-29445Multi-group social network analysis provides comprehensive insights into bonobo grooming behaviorPLOS ONE

Dear Dr. Torfs,

Thank you for submitting your manuscript to PLOS ONE. After careful consideration, we feel that it has merit but does not fully meet PLOS ONE’s publication criteria as it currently stands. Therefore, we invite you to submit a revised version of the manuscript that addresses the points raised during the review process.

We look forward to receiving your revised manuscript.

Kind regards,

Daniel Redhead

Academic Editor

PLOS ONE

Journal Requirements:

Additional Editor Comments:

Two experts in the field, and myself, have now read the manuscript. All of us enjoyed reading the manuscript; it is well-written, clearly structured and speaks to research not only in primatology, but in social network analysis more generally. We all see great potential in the manuscript, with interesting findings among captive bonobos showing that an individual's position within a social network is patterned by age and gender. Both reviewers have flagged some serious issues with the statistical approach applied to analyse the complex data presented in the manuscript. Although Reviewer 2 suggests minor revisions, I must agree with Reviewer 1 as I think major revisions to the analytical strategy are necessary to ensure that reliable conclusions being drawn. The reviews are comprehensive and clearly detail all concerns. I wil not re-state everything here, but will outline the core points that I believe should be addressed during revision:

- Reviewer 1 notes that the many predictions/hypotheses outlined in the introduction should be unpacked and justified in greater detail.

- Use of the many associated centrality metrics should be clearly justified, and potentially re-thought.

- Both reviewers have queried the use of permutations, and this analytical choice should either be clearly justified or re-considered.

- Both reviewers question whether the analysis approach closely reflects the data Reviewer 1 goes further to softly suggest more principled, state-of-the-art analytical strategies (e.g., Hart et al., 2022, Ross et al., 2022), of which I believe would certainly strengthen the manuscript.

- Both reviewers ask for greater clarification of the present analyses and results related to multi-group analysis.

Important references to consider for revision:

Hart, J. D., Weiss, M. N., Franks, D. W., & Brent, L. J. (2021). Bison: A bayesian framework for inference of social networks. bioRxiv.

Hart, J. D., Weiss, M. N., Brent, L. J., & Franks, D. W. (2022). Common permutation methods in animal social network analysis do not control for non-independence. Behavioral ecology and sociobiology, 76(11), 1-10.

Franks, D. W., Weiss, M. N., Silk, M. J., Perryman, R. J., & Croft, D. P. (2021). Calculating effect sizes in animal social network analysis. Methods in Ecology and Evolution, 12(1), 33-41.

Hobson, E. A., Silk, M. J., Fefferman, N. H., Larremore, D. B., Rombach, P., Shai, S., & Pinter‐Wollman, N. (2021). A guide to choosing and implementing reference models for social network analysis. Biological Reviews, 96(6), 2716-2734.

Ross, C. T., McElreath, R., & Redhead, D. (2022). Modelling human and non-human animal network data in R using STRAND. bioRxiv.

Reviewers' comments:

Reviewer's Responses to Questions

**Comments to the Author**

1. Is the manuscript technically sound, and do the data support the conclusions?

Reviewer #1: No

Reviewer #2: Partly

2. Has the statistical analysis been performed appropriately and rigorously? 

Reviewer #1: No

Reviewer #2: Yes

3. Have the authors made all data underlying the findings in their manuscript fully available?

Reviewer #1: No

Reviewer #2: No

4. Is the manuscript presented in an intelligible fashion and written in standard English?

Reviewer #1: Yes

Reviewer #2: Yes

5. Review Comments to the Author

Reviewer #1: I enjoyed reading this paper which presented some interesting data on affiliative networks in captive bonobos. The research reveals some interesting patterns relating sex and age to social network position within these captive groups, and it is great to see this relationship examined across multiple groups of different sizes and composed of different combinations of males and females, and old and young. I thought the framing of the paper in the introduction and discussion was excellent overall. That being said, I am a little concerned by some of the statistical approaches used and while I think there is a good chance these won’t have a substantial affect on the conclusions drawn, it would be good to see more robust methods applied. Thus I have indicated “No” as a response to related questions in the reviewer assessment, although am not a huge fan of these binary indications, as I feel that relatively small adjustments can address many of them.

It would also be really great to see more than just a summary dataset shared here – in my reviewer guide it states “For example, in addition to summary statistics, the data points behind means, medians and variance measures should be available.” And this is not the case with the data shared here, where it could be possible to share edge lists or networks to improve the replicability of these analyses.

One comment on the introduction is that I found the final two paragraphs providing the hypotheses/predictions rather challenging as a “naïve” reader. There are a lot of predictions packed in here and often the rationale for them isn’t especially clear – it hasn’t featured obviously in the introduction and then the explanation comes after the prediction. It may be helpful to play with the structure here to help readers have a good idea of how everything fits together (the table is definitely a big help and more could be made of this).

Building on this, part of the challenge is generated by using five broadly overlapping measures of social centrality, and there are a few related issues here beyond issues with the complexity of the story and elevated risk of false positive results (the latter of which I think is important for the authors to think about). First, how correlated are these different centrality measures in the bonobo networks studied? If some are closely correlated then it perhaps makes them more redundant and exacerbates risks associated with multiple testing. Second, it wasn’t clear to me what affinity would add to the other measures included here as a summary statistic of second order connectivity – it seems the insights here would be explained by any discrepancy between results from strength and eigenvector centrality results. Third, given the small size of the networks studied (and VERY small size of some) I am concerned about the meaningfulness of the indirect measures of centrality (eigenvector centrality and affinity). For the latter in particular there would seem to be some real issues with non-independence of the values of this measure that make it challenging to interpret. But even for eigenvector centrality, what can it tell us in a network of only 3, 4, 5 individuals that strength doesn’t?

Given that the networks vary considerably both in size and sampling intensity (the range in the number of scan samples is sizeable) it would also be good for the authors to consider analytical options that address uncertainty in edge weights and account for this in subsequent analyses - e.g. https://www.biorxiv.org/content/10.1101/2021.12.20.473541v1 - given this is the state-of-the-art this is a softer suggestion but think it would be particularly valuable to work such as this where greater weight could be placed on results from better sampled networks.

Related to this, while I am relatively neutral to the use of permutations in animal social network analysis – in this particular case it is likely that not much is gained from using this approach (see https://doi.org/10.1007/s00265-022-03254-x). However, if the authors stick to using their permutation approach then there are a few additional considerations they need to take care with. First, the choice of permutation approach needs to be carefully justified – it strikes me that the node swaps as structured here work well for the individual-level questions (although perhaps could/should be restricted within networks or networks of similarly sized groups perhaps?) but that the justification for using this approach for the group-level analyses is much less clear. Second, I would strongly advise against combining the permutation approach with subsequent model selection – I would stick to using inference from the full model. Third, it is not consistent to use permutation methods to assess statistical significance because of concerns about non-independence and its affect on standard errors but then use confidence intervals when plotting figures – if permutation tests are the way to go perhaps these should be avoided.

I am also concerned about the structure of the data, and whether the use of random intercepts for individual and group are sufficient to control for this. In particular the fact that some groups are almost identical but differ in one or two members alone would seem likely to cause additional sources of non-independence, especially for relational data, i.e. the different groups are not fully independent of each other. It seems a really tricky dataset, and I’m not quite sure how best to suggest dealing with these types of issue (if they do indeed have the potential to lead to biased results). Regardless, I think more care going into the random effect structure of the model might be important here.

With regard to the group-level analyses, I have several aspects I am confused about. First, given group size is included in the individual-level model it is not immediately clear why a separate analysis for group size is required (albeit the model is more simple). Second (related to the first point), given that group size was already included in the individual-level model and so any inference from (at least the full model) already was with respect to group size effects then it is not clear to me what repeating the analysis after standardising for group size (L320-323) achieves in addition, assuming that the correction is linear? These corrections are also inadequately described (how did you correct), nor are they included in the methods. Finally, with the interpretation of the group size results, this seems to me like it could be non-intuitive, as there may be background (null) relationships between network measure values and group size that depend on aspects of the generative model underlying network structure (e.g. how network density scales with group size). This remains unknown in animal societies but it may be worth reflecting on when discussing these group-level results.

I would also encourage the authors to explore the impact of captivity and the captive environments in their discussion a little more – this does come up (especially in regard to the movement of individuals in a way that mirrors dispersal in the wild), but it feels like there may be other aspects to this. For example, how does group size correspond to the size/complexity of the enclosure and could this impact social structure/affiliative interactions at all? Is there anecdotal evidence of individuals occupying similar positions in different networks or moving individuals causing substantial changes in network structure. I don’t think these make/break interpretations of the results, but they were some questions that arise when thinking about comparing different captive groups.

Line-by-line comments:

L24-25: It would be great if a clearer indication of the direction of results was given here and the two parts of the sentence either separated or more clearly linked with each other.

L40-42: Try and avoid the circularity that arises from the second “network” here – currently the sentence reads as social network analysis is a great tool for analysing social networks, where if the wording was changed it could read that SNA is a great tool for studying social systems and the positions of individuals within them more generally.

L44: Missing citation for the point about longevity

L49: I’m not a primatologist. Is it accurate to say that social bonds are maintained by grooming relationships only or would a more qualified statement be clearer here to avoid potential confusion?

L61: Would suggest providing a more careful and accurate definition of eigenvector centrality. The one here is at odds with the way the eigenvector centrality is often used as a measure and its accuracy will depend on the structure of the network (the one used in the table is much better).

L113-115: This sentence doesn’t make any sense as it is currently written – I think there might be an error which leads to one part seeming to say the opposite to the other?

L120: Again, this may be as a result of my naivety about primatology, but this relies on assumptions about post-reproductive lifespan that haven’t been introduced from the literature yet. Are these post-reproductive lifespans only found in captivity?

L137-141: This hypothesis makes a set of strong assumptions about whether in- and out- interactions are mutually exclusive (presumably not for grooming interactions? You can groom another individual while being groomed yourself?) and the latter part of the argument seems tenuous for strength – I can see the case for increased disparity in large groups but not how this leads to reduced mean in-strength? For example, if one individual spends 60% time of its time being groomed and the rest of individuals no time, then this is the same mean strength as all 6 individuals being groomed for 10% of the time. It strikes me that this argument alone is insufficient to explain differences in mean in-strength.

L165-166: The infants were not included in the networks or not included in the analysis? This is unclear currently.

L171-173: Suggest changing “Given that” to “Because” as initially I misunderstood this sentence because of this initial wording

L178: It is not clear what this reliability score refers to? It is a little concerning if it is a reliability of individual identity for a network study given the impacts this could have on the observed network structure (although perhaps this is the norm in primatology)

Table 3: Would suggest providing this information by individual as well as by data point.

L216-217: Eigenvector centrality was calculated using the weighted networks?

L225-226: It’s not entirely clear to me why the correction is only relevant here?

L235-237: The rationale of using within-group swaps for the questions tackled here is not clearly explained. It makes sense to me (I think) for the individual-based questions, but doesn’t seem like a relevant permutation test for the group-level questions.

L246: I don’t think this is correcting the p-value so much as calculating a new p-value using a different approach.

L248-251: I find these two sentences very unclearly explained and confusing.

Figure 3: How much leverage is there of eigenvector centrality measures from very small groups, especially given the measure isn’t especially meaningful in these very small groups?

L320-323: How have you conducted these standardisations and why is this not in the methods?

L417: Presumably this could also be driven simply by the rules linking group size and network density without invoking any more complex patterns?

L427: How do you propose this is done? It is incredibly challenging, especially when trying to tease apart true biological effects from network effects

I hope my comments help improve the paper.

Reviewer #2: Review of “Multi-group social network analysis provides comprehensive insights into bonobo grooming behavior”

This study examines the relationships between various social network metrics and individual- and group-level traits in bonobos. Overall the manuscript is well-written and logically presented. The findings recapitulate some well-known results, but also extend to a number of novel results. The results will likely be of interest to primatologists interested in social behaviour as well as animal social network practitioners more generally. I have relatively few comments on the manuscript and would be happy to see it published with these changes.

Comments:

Title: The word “comprehensive” in the title suggests that nearly all aspects of bonobo grooming behaviour are examined. I think this is a stretch, and I suspect not exactly what the authors intended. Perhaps they would consider editing the title slightly?

L235: Recent work has shown that permutations don’t account for non-independence in this way (Hart et al., 2022). It doesn’t seem to affect results, so you could just change the justification of using permutations to “as per convention” or something like that.

L245: Very small terminology point - you refer to the distribution of coefficients as a posterior distribution, but since this is a frequentist analysis and is framed as repeated draws of a random variable, it is typically termed a sampling distribution.

L247: Stepwise regression has been critiqued as not respecting a causal model, and potentially leading to inflation of false positives (Smith, 2018). It might be useful to revisit those models, ensure the causal interpretation aligns with your predictions, and if not, rerun appropriate models.

L248 - 250: Could this be rephrased to make it clear that the permutation-based sampling distribution is not representative for small numbers of individuals, hence those small groups were excluded. I think this is point you’re trying to get across, but I struggled to understand what was being said a bit here.

L263 - 264: Adjusting alpha values to include near-significant effects is not necessarily best practice, as a fixed alpha value guarantees a given false positive rate under the null. A focus on confidence/credible intervals might be a better way to go for more nuanced analyses.

L275: Many of the effect sizes in the results sections are reported as being very small (around 0.0001). I assume the scale of the response and predictors is driving this, but perhaps some rescaling could be used to help readers understand the biological implications of these findings.

L438 - 439: I agree that multi-group analyses are important for generalisability, but I don’t quite find any hard evidence in this paper for why that is the case. Perhaps you could tone this message down a bit to make sure this type of statement is fully supported by the findings.

References

Hart, J. D., Weiss, M. N., Brent, L. J., & Franks, D. W. (2022). Common permutation methods in animal social network analysis do not control for non-independence. Behavioral ecology and sociobiology, 76(11), 1-10.

Smith, G. (2018). Step away from stepwise. Journal of Big Data, 5(1), 1-12.

6. PLOS authors have the option to publish the peer review history of their article (what does this mean?). If published, this will include your full peer review and any attached files.

Reviewer #1: No

Reviewer #2: No

---

## [Author Response · Author response to Decision Letter 0]

19 Jan 2023

Response to comments of the editor

Dear Dr. Torfs,

Thank you for submitting your manuscript to PLOS ONE. After careful consideration, we feel that it has merit but does not fully meet PLOS ONE’s publication criteria as it currently stands. Therefore, we invite you to submit a revised version of the manuscript that addresses the points raised during the review process.

If applicable, we recommend that you deposit your laboratory protocols in protocols.io to enhance the reproducibility of your results. Protocols.io assigns your protocol its own identifier (DOI) so that it can be cited independently in the future. For instructions see: https://journals.plos.org/plosone/s/submission-guidelines#loc-laboratory-protocols. 

Additionally, PLOS ONE offers an option for publishing peer-reviewed Lab Protocol articles, which describe protocols hosted on protocols.io. Read more information on sharing protocols at https://plos.org/protocols?utm_medium=editorialemail&utm_source=authorletters&utm_campaign=protocols.

We look forward to receiving your revised manuscript.

Kind regards,

Daniel Redhead

Academic Editor

PLOS ONE

Journal Requirements:

We have revised the PLOS ONE style requirements. Our ethics statement has been provided in the Methods section.

Additional Editor Comments:

Two experts in the field, and myself, have now read the manuscript. All of us enjoyed reading the manuscript; it is well-written, clearly structured and speaks to research not only in primatology, but in social network analysis more generally. We all see great potential in the manuscript, with interesting findings among captive bonobos showing that an individual's position within a social network is patterned by age and gender. Both reviewers have flagged some serious issues with the statistical approach applied to analyse the complex data presented in the manuscript. Although Reviewer 2 suggests minor revisions, I must agree with Reviewer 1 as I think major revisions to the analytical strategy are necessary to ensure that reliable conclusions being drawn. The reviews are comprehensive and clearly detail all concerns. I wil not re-state everything here, but will outline the core points that I believe should be addressed during revision:

• Reviewer 1 notes that the many predictions/hypotheses outlined in the introduction should be unpacked and justified in greater detail.

• Use of the many associated centrality metrics should be clearly justified, and potentially re-thought 

• Both reviewers have queried the use of permutations, and this analytical choice should either be clearly justified or re-considered. 

• Both reviewers question whether the analysis approach closely reflects the data. Reviewer 1 goes further to softly suggest more principled, state-of-the-art analytical strategies (e.g., Hart et al., 2022, Ross et al., 2022), of which I believe would certainly strengthen the manuscript. 

• Both reviewers ask for greater clarification of the present analyses and results related to multi-group analysis. 

We are pleased that the editor enjoyed the manuscript. We have followed the suggestions by all reviewers and the editor and re-analyzed the data, which resulted in minor changes of our results. We have rewritten the paper accordingly. We have also worked on all other major points, as indicated in our comments below. 

Important references to consider for revision:

Hart, J. D., Weiss, M. N., Franks, D. W., & Brent, L. J. (2021). Bison: A bayesian framework for inference of social networks. bioRxiv.

Hart, J. D., Weiss, M. N., Brent, L. J., & Franks, D. W. (2022). Common permutation methods in animal social network analysis do not control for non-independence. Behavioral ecology and sociobiology, 76(11), 1-10.

Franks, D. W., Weiss, M. N., Silk, M. J., Perryman, R. J., & Croft, D. P. (2021). Calculating effect sizes in animal social network analysis. Methods in Ecology and Evolution, 12(1), 33-41.

Hobson, E. A., Silk, M. J., Fefferman, N. H., Larremore, D. B., Rombach, P., Shai, S., & Pinter‐Wollman, N. (2021). A guide to choosing and implementing reference models for social network analysis. Biological Reviews, 96(6), 2716-2734.

Response to Reviewer 1

Reviewer #1: I enjoyed reading this paper which presented some interesting data on affiliative networks in captive bonobos. The research reveals some interesting patterns relating sex and age to social network position within these captive groups, and it is great to see this relationship examined across multiple groups of different sizes and composed of different combinations of males and females, and old and young. I thought the framing of the paper in the introduction and discussion was excellent overall. That being said, I am a little concerned by some of the statistical approaches used and while I think there is a good chance these won’t have a substantial affect on the conclusions drawn, it would be good to see more robust methods applied. Thus I have indicated “No” as a response to related questions in the reviewer assessment, although am not a huge fan of these binary indications, as I feel that relatively small adjustments can address many of them.

We are pleased that Reviewer 1 enjoyed our paper and we thank them for their appreciation and helpful suggestions on how to improve the quality of our manuscript. To address the concerns raised by the two reviewers and the general editor regarding the statistical analysis, we made substantial changes to our analytical approach, stepping away from the node-based permutation tests. The methodological change has led to minor changes to the conclusions drawn and we have rewritten the paper accordingly. 

It would also be really great to see more than just a summary dataset shared here – in my reviewer guide it states “For example, in addition to summary statistics, the data points behind means, medians and variance measures should be available.” And this is not the case with the data shared here, where it could be possible to share edge lists or networks to improve the replicability of these analyses.

Thank you for pointing this out. To allow for more detailed replication of our results we have now added the adjacency matrices with the grooming rates to the supplementary materials (S1 File).

One comment on the introduction is that I found the final two paragraphs providing the hypotheses/predictions rather challenging as a “naïve” reader. There are a lot of predictions packed in here and often the rationale for them isn’t especially clear – it hasn’t featured obviously in the introduction and then the explanation comes after the prediction. It may be helpful to play with the structure here to help readers have a good idea of how everything fits together (the table is definitely a big help and more could be made of this).

We have now expanded our hypothesis section to include a more broad introduction into bonobo grooming behavior. This way, all the necessary rationale and explanations come first, and the final two paragraphs of the introduction are mainly dedicated to listing our different hypotheses.

Building on this, part of the challenge is generated by using five broadly overlapping measures of social centrality, and there are a few related issues here beyond issues with the complexity of the story and elevated risk of false positive results (the latter of which I think is important for the authors to think about). First, how correlated are these different centrality measures in the bonobo networks studied? If some are closely correlated then it perhaps makes them more redundant and exacerbates risks associated with multiple testing. Second, it wasn’t clear to me what affinity would add to the other measures included here as a summary statistic of second order connectivity – it seems the insights here would be explained by any discrepancy between results from strength and eigenvector centrality results. Third, given the small size of the networks studied (and VERY small size of some) I am concerned about the meaningfulness of the indirect measures of centrality (eigenvector centrality and affinity). For the latter in particular there would seem to be some real issues with non-independence of the values of this measure that make it challenging to interpret. But even for eigenvector centrality, what can it tell us in a network of only 3, 4, 5 individuals that strength doesn’t?

We tested for correlation among all social network measures as this could indeed have implications for the interpretation of our results. We found that, even though all network measures correlated with at least one other measure, as could be expected, none of the correlations showed an |r| > 0.70, which was suggested as a threshold by Webber et al. (2020). All social network measures were used in separate models, to avoid violations of statistical assumptions. We also believe that the inclusion of all different measures aids in a more complete understanding of bonobo grooming networks, as they all represent different aspects of grooming behavior. For example, when considering grooming strength, we separated in- from out-strength because we know that grooming is used as a currency for exchange with other commodities like access to food, sex, etc. If we were to combine them, we can no longer make a distinction between grooming efforts made or received. The second reason for us to include so many variables is because some variables represent traditional measures, which allow for comparison with published literature for validation purposes (strength, disparity), while others have not been investigated in so much detail in bonobos and thus allow for novel insights and contributions to current grooming literature (eigenvector centrality, affinity). We have now added a paragraph to the methods section discussing this matter.

L236: “Prior to further analysis, we tested to what extent the grooming network measures correlated with each other. Correlations among different network measures could be high as these are all calculated from the same object (i.e., the grooming matrix), which excludes the use of different network measures in the same model and can influence the interpretation of the results of different models using the social network measure as a response variable [68]. Following the suggestions from Webber et al. [68], we a priori decided to exclude network measures if they showed a correlation coefficient with an absolute value above 0.70 with other network measures. Correlations among all five grooming network measures were assessed using Pearson’s rank correlations in R. We found that all network measures were correlated significantly with at least one other network measure (S2 Table), with the strongest correlation found between out-strength and affinity (r = -0.63, p < 0.001). We decided to investigate all five social network measures further in this study for three reasons: (1) the absolute values of all correlation coefficients were lower than 0.70, (2) all network measures will be used in different models, avoiding violation of statistical assumptions, and (3) each network measure represents a separate aspect of grooming behavior, and therefore, studying the different measures represents a more complete overview of bonobo grooming behavior which will aid in interpreting the results. This also allows for comparison of SNA measures with traditional grooming variables published in literature for validation purposes (such as strength and disparity), while other SNA measures have not been investigated in so much detail in bonobos and thus allow for novel insights and contributions to current grooming literature (such as eigenvector centrality and affinity).”

We agree with the reviewer that in small groups, indirect social network measures might not be highly informative. Still, we believe that the inclusion of the smaller groups is relevant to our study, as it allows for comparison with other studies on similar group sizes, and as it allows for a large variation in group sizes to test for effects of group size on social network position. We have now added a paragraph to the discussion about this matter. In an early stage of this study, we also ran the analysis excluding groups that had less than 5 individuals. This did not have any effect on the results, but since SNA studies on great apes sometimes include groups of 3 or 4 individuals, we decided to include all groups in the final study.

L507: “One important limitation of our study are the small sizes of some of the networks, which could influence indirect social network measures such as eigenvector centrality and affinity. In small networks, these measures are not that informative as these small networks usually show high densities with connections among most, if not all, dyads within the group. However, small networks are inherent to the study of SNA in zoo-housed great apes, as in previous work, group sizes typically ranged from 3 to 14 individuals [15,22,23,58,59]. Therefore, by including networks ranging from 3 to 15 individuals as in our study, we span the variation seen in previous studies which allows for direct comparisons. Moreover, the large range in group sizes allows for broad testing of group size effects on social network position.”

Given that the networks vary considerably both in size and sampling intensity (the range in the number of scan samples is sizeable) it would also be good for the authors to consider analytical options that address uncertainty in edge weights and account for this in subsequent analyses - e.g. https://www.biorxiv.org/content/10.1101/2021.12.20.473541v1 - given this is the state-of-the-art this is a softer suggestion but think it would be particularly valuable to work such as this where greater weight could be placed on results from better sampled networks. Related to this, while I am relatively neutral to the use of permutations in animal social network analysis – in this particular case it is likely that not much is gained from using this approach (see https://doi.org/10.1007/s00265-022-03254-x). However, if the authors stick to using their permutation approach then there are a few additional considerations they need to take care with. First, the choice of permutation approach needs to be carefully justified – it strikes me that the node swaps as structured here work well for the individual-level questions (although perhaps could/should be restricted within networks or networks of similarly sized groups perhaps?) but that the justification for using this approach for the group-level analyses is much less clear. Second, I would strongly advise against combining the permutation approach with subsequent model selection – I would stick to using inference from the full model. Third, it is not consistent to use permutation methods to assess statistical significance because of concerns about non-independence and its affect on standard errors but then use confidence intervals when plotting figures – if permutation tests are the way to go perhaps these should be avoided.

We agree with the reviewer that the use of permutations in social network analysis is debated, with no clear consensus being reached at the moment. Upon suggestion of the reviewer we re-analyzed our data using parametric regressions following Hart et al. 2022. This has resulted in minor changes in results and we have re-written the paper accordingly.

To postulate upon the first point of Reviewer 1, our current data analysis protocol should be robust against differences in sampling intensity. To process our data, we used the “Index Of Interactions” approach, which was proposed by Sosa et al. (2021). As shown in this preprint, the index of interactions actually is robust against observation bias when combined with parametric regression and with permutations, as it returns low rates of false negatives and false positives, even in the presence of observation bias. We have added this to the method section.

L221: “Since the total observation time differed among groups, the grooming matrix was standardized [11] using the ‘index of interactions’ approach [65].”

I am also concerned about the structure of the data, and whether the use of random intercepts for individual and group are sufficient to control for this. In particular the fact that some groups are almost identical but differ in one or two members alone would seem likely to cause additional sources of non-independence, especially for relational data, i.e. the different groups are not fully independent of each other. It seems a really tricky dataset, and I’m not quite sure how best to suggest dealing with these types of issue (if they do indeed have the potential to lead to biased results). Regardless, I think more care going into the random effect structure of the model might be important here.

We agree with the reviewer that not all groups are fully independent of each other, as some of them share certain group members. However, there were only few groups reconsidered where only 1 individual differed between groups (N=3). We had extensive debates about this issue among coauthors but could not reach a consensus on how to better model this using our dataset as every solution that was offered came with its own limitations. For example, one could argue we set up the random structure by nesting ID (individual) into group-period (e.g., Planckendael 2012) into zoo (e.g., Planckendael), to account for the fact that some individuals were observed multiple times in the same group. However, this would not fully reflect the correct structure of group membership in our study population, as some individuals might be in the same zoo but not in the same subgroup (some zoos house multiple separate groups), and as there is also variation among individuals in how long they have been in a group together with other individuals. Also, some individuals might have lived together with certain individuals, got split up, and were then reunited again in another zoo. Such individual experiences are difficult to properly model and cannot be reflected in such a way for a model using datapoints on an individual level. If we were working with dyadic grooming rates, we could add these shared past experiences into the model, but for now, we were not able to do this and acknowledge this limitation in our discussion. We also know from experience that the addition of just one additional female or male (depending on their age and familiarity with the group members) can cause significant disruption within a stable grooming network, which is why we have opted for our current analytical approach. We are also happy to include any suggestions from the reviewer on how to better model these random effects. 

L496: “In our analysis, we considered groups that differed in the presence or absence of at least one group member as fully separate networks, as this could have considerable effects on network structure [22,54] and thus their networks could be considered as independent. Still, different groups with a large proportion of the same group members might not be fully independent of each other, as tenure is known to influence social bonds in bonobos, with stronger social relationships found in individuals that have spent more time together [34,53]. Thus, we might expect that groups sharing certain group members might not be fully independent of each other. However, it is difficult to control for this in our models as we modelled our data on an individual basis. Future studies should look into how variation in dyadic tenures among individual group members structure social networks as a whole, and what the effect of introducing new animals or removing certain group members is on the structure of the social network of the group.”

With regard to the group-level analyses, I have several aspects I am confused about. First, given group size is included in the individual-level model it is not immediately clear why a separate analysis for group size is required (albeit the model is more simple). Second (related to the first point), given that group size was already included in the individual-level model and so any inference from (at least the full model) already was with respect to group size effects then it is not clear to me what repeating the analysis after standardising for group size (L320-323) achieves in addition, assuming that the correction is linear? These corrections are also inadequately described (how did you correct), nor are they included in the methods. Finally, with the interpretation of the group size results, this seems to me like it could be non-intuitive, as there may be background (null) relationships between network measure values and group size that depend on aspects of the generative model underlying network structure (e.g. how network density scales with group size). This remains unknown in animal societies but it may be worth reflecting on when discussing these group-level results.

Since some factors showed significant group size effects in the individual analysis, we simply wanted to test whether any of these associations with sex, age, and rearing history remained after correcting for group sizes or were simply due to group size differences. Our methods for standardization were added to the supplementary materials in the previous version but now we have added a paragraph in the methods section to explain our rationale behind it and to explain how the analysis was performed.

L280: “If an effect of group size was found on a certain network measure, we re-ran the model of this network measure after standardizing it for group size, to test whether any other associations still remained and whether they were independent of the effect of group size. To standardize in-strength and out-strength, we divided the measure by the number of available grooming partners (i.e. N-1, with N being the group size) [7]. Disparity was standardized by calculating the ‘deviation from edge weight disparity’, following the methodology of Kalcher-Sommersguter et al. [23]. Eigenvector centrality and affinity were standardized by scaling the value of each individual to the maximum value within their network [54]. Thus, the maximum in each network has a value of 1 and all others values are scaled towards this value. Subsequent analyses were performed as described above with separate models for each network measure, except that group size was removed as a predictor from all models.”

We have also added the points brought up by the reviewer to our discussion. We now found group size effects on eigenvector centrality and disparity, and especially for eigenvector centrality, these might be the result of scaling effects of the networks. However, in the case of eigenvector centrality, the effect of group size differed between the sexes which implicates that additional social effects are at work.

I would also encourage the authors to explore the impact of captivity and the captive environments in their discussion a little more – this does come up (especially in regard to the movement of individuals in a way that mirrors dispersal in the wild), but it feels like there may be other aspects to this. For example, how does group size correspond to the size/complexity of the enclosure and could this impact social structure/affiliative interactions at all? Is there anecdotal evidence of individuals occupying similar positions in different networks or moving individuals causing substantial changes in network structure. I don’t think these make/break interpretations of the results, but they were some questions that arise when thinking about comparing different captive groups.

This is a good point brought up by the reviewer that we only touched upon lightly in the original manuscript. We have now added a paragraph to the discussion highlighting this matter.

L516: “Ideally, our analyses should be repeated in wild groups of bonobos, as certain aspects of the captive environment might influence grooming rates. For example, we did not account for density (i.e., number of individuals per surface area in the enclosure), even though higher densities could lead to increases in grooming rates [89,90]. In the wild, group size will be independent of available space, and as such, grooming rates and network structure will not depend on any density effects and the social implications of group size could be investigated more reliably.”

Line-by-line comments:

L24-25: It would be great if a clearer indication of the direction of results was given here and the two parts of the sentence either separated or more clearly linked with each other.

We expanded the abstract such that we could give more attention to the results.

L23: “Our results showed age-effects on all investigated measures: for females, all measures except for affinity showed quadratic relationships with age, while in males, the effects of age were more variable depending on the network measure. Bonobos with atypical rearing histories showed lower out-strength, while in-strength was only impacted by rearing history in males. Group size showed a negative association with disparity and eigenvector centrality, while sex ratio did not influence any of the investigated measures.”

L40-42: Try and avoid the circularity that arises from the second “network” here – currently the sentence reads as social network analysis is a great tool for analysing social networks, where if the wording was changed it could read that SNA is a great tool for studying social systems and the positions of individuals within them more generally.

We changed the wording accordingly.

L41: “Social network analysis (SNA) provides a useful toolkit to study social systems and an individual’s social position within the group, as it offers an opportunity to quantify various measures that represent different aspects of these direct and indirect relationships [2,7,8].”

L44: Missing citation for the point about longevity

We added the citation.

L49: I’m not a primatologist. Is it accurate to say that social bonds are maintained by grooming relationships only or would a more qualified statement be clearer here to avoid potential confusion?

We have rephrased this sentence since social bonds do not solely depend on grooming relationships, but grooming relationships are a good proxy for social bonds and are often used to study sociality in primates. 

L50: “In primates, the time spent grooming others is often used as a proxy for the strength of their social bonds [13,14].”

L61: Would suggest providing a more careful and accurate definition of eigenvector centrality. The one here is at odds with the way the eigenvector centrality is often used as a measure and its accuracy will depend on the structure of the network (the one used in the table is much better).

We rephrased this sentence to include a more accurate definition of eigenvector centrality.

L62: “eigenvector centrality represents an individual’s embeddedness within the entire grooming network as it is a measure for how well an individual is associated, as well as how well their connections are associated themselves”

L113-115: This sentence doesn’t make any sense as it is currently written – I think there might be an error which leads to one part seeming to say the opposite to the other?

This sentence indeed contained an error. It has been rewritten correctly. 

L137: “We predict males will have higher affinity compared to females, because males tend to prefer females as grooming partners [40–43,45], and usually occupy the lowest ranks in the hierarchy and might therefore focus their grooming on high-ranking individuals that are expected to have high strengths [31,47].”

L120: Again, this may be as a result of my naivety about primatology, but this relies on assumptions about post-reproductive lifespan that haven’t been introduced from the literature yet. Are these post-reproductive lifespans only found in captivity?

We have now added a statement about the age span of the reproductive period in bonobos.

L115: “thus grooming activity is expected to be highest during the time-frame of reproduction, which in female bonobos lies roughly between 6 and 40 years old [49].”

L137-141: This hypothesis makes a set of strong assumptions about whether in- and out- interactions are mutually exclusive (presumably not for grooming interactions? You can groom another individual while being groomed yourself?) and the latter part of the argument seems tenuous for strength – I can see the case for increased disparity in large groups but not how this leads to reduced mean in-strength? For example, if one individual spends 60% time of its time being groomed and the rest of individuals no time, then this is the same mean strength as all 6 individuals being groomed for 10% of the time. It strikes me that this argument alone is insufficient to explain differences in mean in-strength.

We agree with the reviewer that our hypotheses regarding the link between in-strength and group size needed reconsideration. We have adjusted our hypotheses to say that in-strength shows no changes with group size, but will rather be dependent on individual ‘popularity’ and social attractiveness, which in turn depends on individual characteristics like sex, age, and rearing history.

L152: “Group size is not expected to impact in-strength as the amount of grooming received will rather depend on individual characteristics such as sex, age, and rearing history.”

L165-166: The infants were not included in the networks or not included in the analysis? This is unclear currently.

Juveniles were not included in the networks, as we had no detailed data on these individuals for all observation periods. This is stated more clearly now.

L181: “All groups consisted of individuals of both sexes, and some groups also contained juveniles and/or infants (i.e., individuals under 7 years old). Juveniles and infants were not included in the networks as we did not collect reliable data on their grooming behavior.”

L171-173: Suggest changing “Given that” to “Because” as initially I misunderstood this sentence because of this initial wording

We changed the wording accordingly: 

L187: “Because some individuals were transferred between groups over time, and in some institutions groups were observed multiple times over the 11 year time-span, certain individuals were observed more than once.”

L178: It is not clear what this reliability score refers to? It is a little concerning if it is a reliability of individual identity for a network study given the impacts this could have on the observed network structure (although perhaps this is the norm in primatology)

This inter-observer reliability score refers to the similarity between observers in observing and coding the correct behaviors shown in the videos using our standardized ethogram. Small differences are typically found between observers in the speed of coding which result in slightly divergent durations and frequencies of mostly fast-occurring behaviors like scratching. This thus does not have anything to do with individual recognition of animals. All observers go through an extensive training period and are required to be able to identify each individual without error before they can start their data collection. 

Table 3: Would suggest providing this information by individual as well as by data point.

We now provide the information by individual between brackets in the same table.

L216-217: Eigenvector centrality was calculated using the weighted networks?

Indeed, all measures were calculated using the weighted network matrices. We have now stated this more clearly.

L232: “All measures were calculated from the weighted networks.”

L225-226: It’s not entirely clear to me why the correction is only relevant here?

Since we changed our analytic approach, the original sentence was deleted as we could now use specialized R-packages to perform post-hoc comparisons.

L235-237: The rationale of using within-group swaps for the questions tackled here is not clearly explained. It makes sense to me (I think) for the individual-based questions, but doesn’t seem like a relevant permutation test for the group-level questions. L246: I don’t think this is correcting the p-value so much as calculating a new p-value using a different approach. L248-251: I find these two sentences very unclearly explained and confusing. 

As we have changed our analytical approach, these original sentences are now deleted from the manuscript.

Figure 3: How much leverage is there of eigenvector centrality measures from very small groups, especially given the measure isn’t especially meaningful in these very small groups?

Their leverage is minimal as there are only a few datapoints available from these small groups. We also added a paragraph to the discussion about the caveats associated with small groups (see above).

L320-323: How have you conducted these standardisations and why is this not in the methods?

A paragraph has now been added to the methods section (see also above).

L417: Presumably this could also be driven simply by the rules linking group size and network density without invoking any more complex patterns?

This might indeed be the case. We also based our predictions on these rules. We have now more explicitly stated this again in our discussion. Still, it remains interesting that the effect differs between males and females, suggesting that certain sex-differences in sociality might influence the response of eigenvector centrality to group size, so we still kept this in the discussion.

L474: “In line with our expectations, eigenvector centrality significantly decreased with group size. Larger groups have less dense and more modular networks [28], leading to lower eigenvector centralities for individual group members. However, it is interesting that the effect differed between the sexes, with a larger effect in females than in males. Females show a greater reduction in eigenvector centrality with group size than males do, which could be explained by differences in social patterns between the sexes. Females will form strong grooming bonds with all group members, while male bonobos are more selective in their grooming bonds and tend to prefer high-strength female group members to groom with. Hence, when group size increases and groups become more female-biased in our sample, females will more drastically drop in centrality as more equally central females are present, while males have more high-strength females to associate with, thereby showing a less drastic reduction in centrality. Moreover, the finding that group size effects on eigenvector centrality differ between males and females illustrate why multi-group studies are important. As seen in Fig 3B, in smaller groups, females tend to have higher eigenvector centrality than males, while in the larger groups, males seem to have higher eigenvector centrality than females. If only one group is studied, a sex-difference in eigenvector centrality might actually simply be a consequence of the group size and might not reflect an absolute difference in social embeddedness between the sexes.”

L427: How do you propose this is done? It is incredibly challenging, especially when trying to tease apart true biological effects from network effects

Based on research in previous publications (e.g., Balasubramaniam et al., 2021; Crailsheim et al., 2020; Kalcher-Sommersguter et al., 2015; Kasper & Voelkl, 2009), we have standardized our measures and re-run the analysis. We have now added our methods and the rationale behind it more clearly in the methods section of the manuscript (see above).

I hope my comments help improve the paper.

We would like to thank this reviewer again for their careful review of our paper and helpful comments for improvement of our manuscript.

Response to Reviewer 2

Reviewer #2: Review of “Multi-group social network analysis provides comprehensive insights into bonobo grooming behavior”

This study examines the relationships between various social network metrics and individual- and group-level traits in bonobos. Overall the manuscript is well-written and logically presented. The findings recapitulate some well-known results, but also extend to a number of novel results. The results will likely be of interest to primatologists interested in social behaviour as well as animal social network practitioners more generally. I have relatively few comments on the manuscript and would be happy to see it published with these changes.

We thank the reviewer for their kind and helpful feedback on our manuscript. Based on the feedback from this reviewer and also from reviewer 1 we have decided to opt for a different statistical approach which hopefully alleviated most of the concerns raised below. 

Comments:

Title: The word “comprehensive” in the title suggests that nearly all aspects of bonobo grooming behaviour are examined. I think this is a stretch, and I suspect not exactly what the authors intended. Perhaps they would consider editing the title slightly?

We changed the title to: “Multi-group analysis of grooming networks in a highly social primate”.

L235: Recent work has shown that permutations don’t account for non-independence in this way (Hart et al., 2022). It doesn’t seem to affect results, so you could just change the justification of using permutations to “as per convention” or something like that.

We did decide to change our analysis to parametric regression, as suggested also by Reviewer 1. It indeed only had minor effects on the results and final conclusions.

L245: Very small terminology point - you refer to the distribution of coefficients as a posterior distribution, but since this is a frequentist analysis and is framed as repeated draws of a random variable, it is typically termed a sampling distribution. L248 - 250: Could this be rephrased to make it clear that the permutation-based sampling distribution is not representative for small numbers of individuals, hence those small groups were excluded. I think this is point you’re trying to get across, but I struggled to understand what was being said a bit here.

As we have changed our analytical approach, the original sentences were not relevant anymore and are now deleted from the manuscript.

L247: Stepwise regression has been critiqued as not respecting a causal model, and potentially leading to inflation of false positives (Smith, 2018). It might be useful to revisit those models, ensure the causal interpretation aligns with your predictions, and if not, rerun appropriate models.

Due to changes in our analytical approach we now select the best-fitting model using AIC.

L263 - 264: Adjusting alpha values to include near-significant effects is not necessarily best practice, as a fixed alpha value guarantees a given false positive rate under the null. A focus on confidence/credible intervals might be a better way to go for more nuanced analyses.

While we agree with the reviewer on this comment, it is unfortunately not possible to work with confidence intervals when analyzing linearity of our dominance data using the available software as it only returns the value of linearity and a p-value based on permutations. While David’s scores are the preferred method for determining dominance and are in our opinion more reliable than, for example, the use of keeper interpretations of dominance (which are also commonly used in literature), they have the limitation that they require data on submissive interactions within the majority of dyads in the group to produce significant results. However, in reality, submissive behavior and aggression are rare in our populations and definitely less frequent and even sometimes absent in female-female dyads. With higher uncertainties (e.g., dyads with no interactions) in the flee-upon-aggression matrix, it is difficult to obtain a significant linear dominance hierarchy because some dyads will occupy similar rank positions. However, if we were to apply a more stringent approach (e.g., alpha set at 0.05), we would have only 2 groups left which does not allow for a representation of sex-specific rank correlations with age. With this limitation in mind we decided not to include our rank results into our SNA models, but rather use them as a visual aid to help interpret the age-related changes in grooming. 

L275: Many of the effect sizes in the results sections are reported as being very small (around 0.0001). I assume the scale of the response and predictors is driving this, but perhaps some rescaling could be used to help readers understand the biological implications of these findings.

These small effect sizes are indeed the consequence of the scale of the response variables and predictors. We chose not to rescale our variables as our data is easy to interpret in terms of percentages, e.g. out-strength refers to the frequency of grooming given, with an out-strength of 0.15 on a graph indicating that this individual groomed 15% of the time during observations.

L438 - 439: I agree that multi-group analyses are important for generalisability, but I don’t quite find any hard evidence in this paper for why that is the case. Perhaps you could tone this message down a bit to make sure this type of statement is fully supported by the findings.

Thank you for pointing this out. While we did not explicitly mention this in great detail in the manuscript, our findings regarding the sex-group size interaction on eigenvector centrality offer a clear example of the importance of multi-group studies. When comparing sex-effects on eigenvector centrality in the different groups on fig 3B, we see that in smaller groups it is the females who have higher centrality whereas in larger groups males have a tendency towards higher centrality. If any single group would have been studied alone, the large variation and opposite effects based on group size would not be detected. We have added this to the discussion.

L484: “Moreover, the finding that group size effects on eigenvector centrality differ between males and females illustrate why multi-group studies are important. As seen in Fig 3B, in smaller groups, females tend to have higher eigenvector centrality than males, while in the larger groups, males seem to have higher eigenvector centrality than females. If only one group is studied, a sex-difference in eigenvector centrality might actually simply be a consequence of the group size and might not reflect an absolute difference in social embeddedness between the sexes.”

References

Hart, J. D., Weiss, M. N., Brent, L. J., & Franks, D. W. (2022). Common permutation methods in animal social network analysis do not control for non-independence. Behavioral ecology and sociobiology, 76(11), 1-10.

Smith, G. (2018). Step away from stepwise. Journal of Big Data, 5(1), 1-12.

---

## [Decision Letter · Decision Letter 1]

12 Mar 2023

PONE-D-22-29445R1Multi-group analysis of grooming network position in a highly social primatePLOS ONE

Dear Dr. Torfs,

Thank you for submitting your manuscript to PLOS ONE. After careful consideration, we feel that it has merit but does not fully meet PLOS ONE’s publication criteria as it currently stands. Therefore, we invite you to submit a revised version of the manuscript that addresses the points raised during the review process.

 Both reviewers and I are pleased to see many of our suggested changes have been made, and believe that these changes have greatly strengthened the manuscript. For this reason, I suggest only minor changes need to be made to the manuscript to make it suitable for publication. That being said, Reviewer 1 raises several important points about the analyses presented in the manuscript, which should be addressed in detail in your revision.  These comments are detailed below. 

We look forward to receiving your revised manuscript.

Kind regards,

Daniel Redhead

Academic Editor

PLOS ONE

Journal Requirements:

Additional Editor Comments (if provided):

The manuscript is reading much better now, and I have slightly more confidence in the analyses that are presented. I agree with Reviewer 1 on several core concerns that remain in the current manuscript:

The use of indirect metrics of centrality in small networks. Is it meaningful to interpret such a metric such as eigenvector centrality on a network of such a size (e.g., 3 nodes/individuals)? Given how such metrics are calculated, I would suggest that it is not and that only analyses of direct measures should be included.Why did the authors not opt for more state-of-the-art approaches  (e.g., using a social relations model) highlighted by Reviewer 1 and myself? This analytical choice should be at least acknowledged and justified in the manuscript. Please rethink and at least better justify the use of David scores. Please include confidence intervals for all relevant results and further respond to Reviewer 2's points about interpretation of coefficients. 

Reviewers' comments:

Reviewer's Responses to Questions

**Comments to the Author**

1. If the authors have adequately addressed your comments raised in a previous round of review and you feel that this manuscript is now acceptable for publication, you may indicate that here to bypass the “Comments to the Author” section, enter your conflict of interest statement in the “Confidential to Editor” section, and submit your "Accept" recommendation.

Reviewer #1: (No Response)

Reviewer #2: (No Response)

2. Is the manuscript technically sound, and do the data support the conclusions?

Reviewer #1: Yes

Reviewer #2: Yes

3. Has the statistical analysis been performed appropriately and rigorously? 

Reviewer #1: Yes

Reviewer #2: Yes

4. Have the authors made all data underlying the findings in their manuscript fully available?

Reviewer #1: Yes

Reviewer #2: Yes

5. Is the manuscript presented in an intelligible fashion and written in standard English?

Reviewer #1: Yes

Reviewer #2: Yes

6. Review Comments to the Author

Reviewer #1: In general, I feel the authors have done a great job addressing concerns from the reviewers, and the paper is now clear, well-written and well-explained.

I have to admit that I remain unconvinced about the wisdom of using indirect measures in the smallest networks presented here. I appreciate the authors have provided a detailed caveat in the discussion, and perhaps this is sufficient. However, this remains problematic if the measures used have different biological interpretations in these very small networks. It is unclear to me what variation in eigenvector centrality would mean in such small networks, and I am concerned that variation in affinity in small networks represents a different trait to its variation in larger networks given that in networks of 3 individuals the least connected individual will almost certainly have the highest affinity which is not true in larger networks (e.g. in toy small networks with 3 nodes, affinity often seems highest for the least integrated individual). Given the results seem consistent when the smallest networks are excluded (mentioned by the authors in their response), I am happy to leave it to the discretion of the editor as to how important this concern is.

It is also a little disappointing to see the authors not try some of the more state-of-the-art statistical approaches highlighted by the editor, although I think that given the nature of the sampling here (fairly comprehensive and relatively even across groups), the current approach is probably adequate, and I wouldn’t expect the results to change substantially. I do think the current approach selected has improved the clarity of the paper considerably. (A quick side note for the authors to consider for their future research. While the index of interactions approach is probably OK in this case given the intensity of data collection and relative ease with which individuals can be observed, it is not accurate to say that it deals as well with variation in sampling intensity as newer approaches. Through being an index, individuals observed grooming each other 1 out of 2 times is recorded in the same way as individuals recorded grooming each other 100 out of 200 times (index of 0.5). However, there is a huge difference in the amount of information these two sets of observations carry about the social relationship between two individuals, which is why approaches such as the BISoN framework recommended previously, and other similar approaches are preferable in general)

I am also not fully convinced by how the authors have used the AIC approach here for model selection. First, I’m not sure it fully addresses the other reviewer’s concerns related to the statistical modelling (which seemed more aimed at pushing for a causal understanding, and would perhaps favour a well justified full model inference approach). Second, the mixture of AIC model selection (which weighs the support for the model as a whole) and F tests that test the importance of single variables within the model seems rather inconsistent.

L311-318: As to some extent already highlighted by the authors, there are severe limitations to this analysis – the authors extract David scores without accounting for uncertainty around them, the complexity of the fixed effect structure (number of variables plus interactions) is high for a small sample size (n=40), and the number of random effect levels is also small (n=5) to accurately estimate random effect variance. I’m a little torn as the model results remain fairly convincing despite (or perhaps in the case of the uncertainty) because of these limitations, but equally I wonder whether the plot (Fig. 1F) is suitably convincing without needing to resort to using the statistical approach in the first place. Again, I am happy to leave this to the discretion of the editor/authors.

Otherwise, I have just a few very minor points.

L181-182: There is a typo in this sentence

L332-333: It is not clear to me how an “exponential” decline in affinity with age was detected when the relationship(s) tested were quadratic (and linear). I assume the authors were trying to draw attention to the fact that affinity only declined in older individuals, and the rate of decline accelerated with age (Fig. 1d)?

On a related point, there seems to be a missing line/confidence interval Fig. 1d? It would also be good to position the legend (male/female colours) in such a way that it applies to all panels of the plot.

L400-405: There are some potential links here with wider literature on social selectivity with age (e.g. https://doi.org/10.1073/pnas.2209180119 and the papers it cites).

L450: Describing this as a “vast” majority seems excessive. It would also be good to point to the fact you only have 22 groups so have relatively low power at this level of the analysis.

L478-481: I find this explanation hard to understand as it rests on sex ratios influencing centrality measures, and you did not find a direct effect of these measures when they were included in your statistical models

L502: effectS

Reviewer #2: I’m pleased to see this manuscript again, and I’m glad the authors have made many of the changes suggested by the reviewers. I feel the manuscript is now much improved. I am now mostly happy with the manuscript, though I still have two outstanding concerns.

Firstly, I thank the authors for their response regarding my comment about the scale of the predictors, and how that might affect the interpretability of the regression coefficients. The authors note that they feel the scale is appropriate because it allows coefficients to be interpreted as changes in % time spent. I agree with the authors in theory, but in practice, a coefficient value of e.g. β = -0.002 (from L331) is difficult to interpret biologically. Perhaps the authors could also provide a short explanation of what this means biologically, for example, an difference in age of 10 years would be associated with a difference in out-strength of around 2%. This might help readers to appreciate the biological magnitude of these effects.

A related point is that the authors were unable to report confidence intervals for their models. I strongly feel reporting intervals helps to biologically ground the results, so that readers appreciate the scientific implications of the findings. Confidence intervals can usually be extracted from most popular software packages. I understand lmerTest runs tests on lme4 models, and lme4 supports the computation of confidence intervals. I would encourage the authors to revisit this point and see if intervals could be reported. I feel it would really help readers to appreciate these interesting findings.

I’m looking forward to seeing the paper published after these changes are made.

7. PLOS authors have the option to publish the peer review history of their article (what does this mean?). If published, this will include your full peer review and any attached files.

Reviewer #1: No

Reviewer #2: No

---

## [Author Response · Author response to Decision Letter 1]

16 Mar 2023

Response to comments of the editor

Dear Dr. Torfs,

Thank you for submitting your manuscript to PLOS ONE. After careful consideration, we feel that it has merit but does not fully meet PLOS ONE’s publication criteria as it currently stands. Therefore, we invite you to submit a revised version of the manuscript that addresses the points raised during the review process.

Both reviewers and I are pleased to see many of our suggested changes have been made, and believe that these changes have greatly strengthened the manuscript. For this reason, I suggest only minor changes need to be made to the manuscript to make it suitable for publication. That being said, Reviewer 1 raises several important points about the analyses presented in the manuscript, which should be addressed in detail in your revision. These comments are detailed below. 

We look forward to receiving your revised manuscript.

Kind regards,

Daniel Redhead

Academic Editor

PLOS ONE

Journal Requirements:

We have reviewed the reference list. No retracted papers were included in our study.

Additional Editor Comments (if provided):

The manuscript is reading much better now, and I have slightly more confidence in the analyses that are presented. I agree with Reviewer 1 on several core concerns that remain in the current manuscript:

• The use of indirect metrics of centrality in small networks. Is it meaningful to interpret such a metric such as eigenvector centrality on a network of such a size (e.g., 3 nodes/individuals)? Given how such metrics are calculated, I would suggest that it is not and that only analyses of direct measures should be included.

• Why did the authors not opt for more state-of-the-art approaches (e.g., using a social relations model) highlighted by Reviewer 1 and myself? This analytical choice should be at least acknowledged and justified in the manuscript. 

• Please rethink and at least better justify the use of David scores. 

• Please include confidence intervals for all relevant results and further respond to Reviewer 2's points about interpretation of coefficients. 

We are pleased that the editor enjoyed this new version of the manuscript. We have followed the suggestions of all reviewers and the editor and included responses to each concern in more detail below. We would like to thank the editor again for their helpful comments on this manuscript, which greatly improved the study as a whole.

Response to Reviewer 1

Reviewer #1: In general, I feel the authors have done a great job addressing concerns from the reviewers, and the paper is now clear, well-written and well-explained.

We are pleased that Reviewer 1 enjoyed the new version of our manuscript. 

I have to admit that I remain unconvinced about the wisdom of using indirect measures in the smallest networks presented here. I appreciate the authors have provided a detailed caveat in the discussion, and perhaps this is sufficient. However, this remains problematic if the measures used have different biological interpretations in these very small networks. It is unclear to me what variation in eigenvector centrality would mean in such small networks, and I am concerned that variation in affinity in small networks represents a different trait to its variation in larger networks given that in networks of 3 individuals the least connected individual will almost certainly have the highest affinity which is not true in larger networks (e.g. in toy small networks with 3 nodes, affinity often seems highest for the least integrated individual). Given the results seem consistent when the smallest networks are excluded (mentioned by the authors in their response), I am happy to leave it to the discretion of the editor as to how important this concern is.

To circumvent the issues related to indirect network measures in small groups, we have limited our analyses of affinity and eigenvector centrality to groups with 5 or more individuals, only slightly reducing the size of the dataset compared to the direct measures (10 individuals from 3 groups were removed for the analyses of indirect measures). We chose this threshold of a group size of 5 to remove the smallest groups from our dataset but also minimize the number of groups that had to be removed, such that enough power remained to test for group-level effects on the indirect measures. We decided to re-analyze affinity and eigenvector centrality in our manuscript, rather them remove them completely, as these are the most novel within the bonobo literature, since only two small single-group studies have previously reported on eigenvector centrality and none have reported on affinity. Hence, we believe that it is of interest to maintain these indirect grooming network measures in the manuscript, and we hope that we have solved the concerns of Reviewer 1 and the editor by running the analyses on a smaller dataset excluding these small groups. For affinity, the results remained the same as in the previous version of the manuscript. For eigenvector centrality, rearing-history was a fixed effect in the remaining best model whereas in the full dataset this was not the case. Parent-reared bonobos are more central in the network compared to other-reared bonobos, which was one of our hypotheses. We have rewritten our results and discussion section accordingly. We hope that our re-analysis resolved the concerns raised by Reviewer 1 and the editor.

It is also a little disappointing to see the authors not try some of the more state-of-the-art statistical approaches highlighted by the editor, although I think that given the nature of the sampling here (fairly comprehensive and relatively even across groups), the current approach is probably adequate, and I wouldn’t expect the results to change substantially. I do think the current approach selected has improved the clarity of the paper considerably. (A quick side note for the authors to consider for their future research. While the index of interactions approach is probably OK in this case given the intensity of data collection and relative ease with which individuals can be observed, it is not accurate to say that it deals as well with variation in sampling intensity as newer approaches. Through being an index, individuals observed grooming each other 1 out of 2 times is recorded in the same way as individuals recorded grooming each other 100 out of 200 times (index of 0.5). However, there is a huge difference in the amount of information these two sets of observations carry about the social relationship between two individuals, which is why approaches such as the BISoN framework recommended previously, and other similar approaches are preferable in general)

We apologize for not meeting some of the expectations of the Reviewer, but we were under the impression that it mostly concerned the removal of our permutation approach, which we implemented in the novel version of the manuscript. We were reluctant to implement the BISoN package at this stage of our research as it is, to our knowledge, currently only available as a non-peer reviewed preprint and has not been used in other SNA studies. Given the current finalized state of this paper, we were therefore hoping to include the BISoN package in a follow-up study where we want to link SNA data to physiological measures. We have contacted the author of the package with a request for assistance with these analyses since we are running into some issues regarding certain limitations of the bisonR package. We hope the reviewer and editor both agree that the field of SNA is a rapidly evolving one with lots of opposing views regarding methodological approaches, but we hope that given the size and data collection protocol of our dataset the current analytical approach is considered sufficient and offers comparable results to what is currently the norm for published primate SNA research. 

We have added a paragraph to the discussion where we discuss the usefulness of novel techniques in improving the reliability of social network analyses, hereby highlighting their existence and potential for application in future SNA studies.

L537-550: “As the study of animal social networks is a rapidly developing field, especially regarding advances in statistical methodology, future studies on primate grooming networks could aim to include more state-of-the-art analytical approaches to investigate variation in grooming network measures. For example, besides using parametric models or permutation methods, generative models can be used to study social network data as well, as they account for the non-independence of social network data [89]. In addition, as sampling effort varies across groups, but also within groups (e.g., certain dyads might be observed more often than others and hence their grooming relationship might be more ‘certain’), uncertainty in network edges could be taken into account in downstream analyses. Instead of correcting for variation in sampling effort using commonly used social network indices, using other frameworks such as the newly developed BISoN framework [90] could provide a way to incorporate sampling effort and edge uncertainty into down-stream analyses. In our current study, sampling effort was relatively even across groups and across dyads within the groups due to the high visibility of individuals, reducing edge uncertainty. Nonetheless, future studies could take advantage of these newly developing analytical techniques to more reliably investigate primate social networks.”

I am also not fully convinced by how the authors have used the AIC approach here for model selection. First, I’m not sure it fully addresses the other reviewer’s concerns related to the statistical modelling (which seemed more aimed at pushing for a causal understanding, and would perhaps favour a well justified full model inference approach). Second, the mixture of AIC model selection (which weighs the support for the model as a whole) and F tests that test the importance of single variables within the model seems rather inconsistent.

We are not entirely sure we understand what the reviewer means with this remark. Based on the comment of the other reviewer we now apply AIC as a model selection approach rather than a backward model selection approach and then report the variables included in this best model with their estimates and significance in the model. We did make an error on our end by reporting F-tests, since we did not run additional F-tests to determine significance of factors but rather rely on estimates and corresponding test statistics of the fixed effects and interactions (if any were present in the final model) from the model summaries implemented within the lmerTest package, which uses t-tests. We are happy to incorporate a different approach if needed, but would like more feedback on how to do so. 

L275-278: “We first ran full models and then reduced the models based on the Akaike Information Criterium (AIC), such that the most parsimonious model with the lowest AIC value remained [72]. Model summaries were used to determine the parametric coefficients of the fixed effects in the remaining models.”

L311-318: As to some extent already highlighted by the authors, there are severe limitations to this analysis – the authors extract David scores without accounting for uncertainty around them, the complexity of the fixed effect structure (number of variables plus interactions) is high for a small sample size (n=40), and the number of random effect levels is also small (n=5) to accurately estimate random effect variance. I’m a little torn as the model results remain fairly convincing despite (or perhaps in the case of the uncertainty) because of these limitations, but equally I wonder whether the plot (Fig. 1F) is suitably convincing without needing to resort to using the statistical approach in the first place. Again, I am happy to leave this to the discretion of the editor/authors.

We agree that our analysis of David’s scores were not highly reliable, which is why we decided to not incorporate rank into the models but rather use it as a visual aid to explain the age-related patterns that many network measures follow in our study. As the rank analysis is not the focus of our study, there is no real need to run statistical analyses to model age effects on them so following the reviewers suggestions we removed them from the study and only included the visualization to support our discussion.

Otherwise, I have just a few very minor points.

L181-182: There is a typo in this sentence

The typo has been corrected:

L181-182: “Juveniles and infants were not included in the networks as we did not collect reliable data on their grooming behavior.”

L332-333: It is not clear to me how an “exponential” decline in affinity with age was detected when the relationship(s) tested were quadratic (and linear). I assume the authors were trying to draw attention to the fact that affinity only declined in older individuals, and the rate of decline accelerated with age (Fig. 1d)? On a related point, there seems to be a missing line/confidence interval Fig. 1d? It would also be good to position the legend (male/female colours) in such a way that it applies to all panels of the plot.

We indeed meant to refer to a quadratic relationship, rather than an exponential one. We have rewritten this sentence accordingly:

L337-339: “Affinity showed a tendency towards a quadratic relationship with age in male bonobos, with the rate of decline being shallow in younger bonobos, but accelerating with age, with the lowest values found in old male bonobos.”

We re-made the figures and placed a legend in each panel where sex-differences in age patterns were found. We also added the missing trendline in panel 1D.

L400-405: There are some potential links here with wider literature on social selectivity with age (e.g. https://doi.org/10.1073/pnas.2209180119 and the papers it cites).

Here, the reviewer refers to our finding that old male bonobos groom less than younger males (the same goes for females), as shown by a significant reduction in their out-strength. This does not necessarily reflect a change in social selectivity, as these males might still invest equally in different grooming partners, but might just show a general decrease in grooming efforts. But we included this suggestion later on, where we discuss disparity, given that disparity increases in older bonobos, which is in line with higher social selectivity. We have added this paper to the discussion there as we only referred to chimpanzee studies in the previous version of the manuscript, but not to studies investigating other primate species.

L445-453: “With age, disparity declined, but in old age, disparity increased again, showing bonobos become more selective again after their thirties. These results are in line with a recent body of literature reporting similar effects of higher social selectivity with age across a variety of primate species [e.g., 19,51,85].”

L450: Describing this as a “vast” majority seems excessive. It would also be good to point to the fact you only have 22 groups so have relatively low power at this level of the analysis.

We have rewritten our discussion here following these suggestions:

L471-481: “However, this result is to be treated with caution as the majority of groups in our study were female-biased (14/22 female-biased groups versus 3/22 being male-biased), and there was a confound between group size and sex ratio, with the largest groups in our study all being female-biased (S1 File and S1 Table). This is important to consider when interpreting the results, as for some network measures, opposite effects between group size and sex-ratio would be expected in large male-biased groups (Table 2), although these are rare in bonobos due to the lower life-expectancy of male bonobos [49,87]. Moreover, due to the relatively low power our study has on the group-level, a possible effect of sex ratio on grooming network position might have gone undetected. Still, it needs to be noted that a previous study did not find any effect of group sex ratio on grooming patterns in wild bonobos [88], so perhaps bonobo grooming strategies are relatively independent of the sex ratio of the group.”

L478-481: I find this explanation hard to understand as it rests on sex ratios influencing centrality measures, and you did not find a direct effect of these measures when they were included in your statistical models

We agree with the reviewer here that this sentence was not entirely clear. We have now rewritten the sentence as follows:

L498-505: “Females show a greater reduction in eigenvector centrality with group size than males do, which could be explained by differences in social patterns between the sexes. Females will form strong grooming bonds with all group members, while male bonobos are more selective in their grooming bonds and tend to prefer high-strength female group members to groom with. Hence, when group size increases and thus the absolute number of females in the group increases, females will more drastically drop in centrality as more equally central females are present, while males will have more high-strength females available to associate with, thereby showing a less drastic reduction in centrality.”

L502: effectS

We have corrected this typo:

L513: “As the effects of our individual level variables (sex and age) on grooming metrics remained identical after correction for group size, they appear to have a robust effect on grooming behavior that is independent of demographic variation.”

We would like to thank the reviewer for their helpful and constructive comments on the current and previous version of our manuscript.

Response to Reviewer 2

I’m pleased to see this manuscript again, and I’m glad the authors have made many of the changes suggested by the reviewers. I feel the manuscript is now much improved. I am now mostly happy with the manuscript, though I still have two outstanding concerns.

We are pleased that Reviewer 2 enjoyed the new version of our manuscript and that they acknowledge the improvement of the study as a whole.

Firstly, I thank the authors for their response regarding my comment about the scale of the predictors, and how that might affect the interpretability of the regression coefficients. The authors note that they feel the scale is appropriate because it allows coefficients to be interpreted as changes in % time spent. I agree with the authors in theory, but in practice, a coefficient value of e.g. β = -0.002 (from L331) is difficult to interpret biologically. Perhaps the authors could also provide a short explanation of what this means biologically, for example, an difference in age of 10 years would be associated with a difference in out-strength of around 2%. This might help readers to appreciate the biological magnitude of these effects.

We understand why reviewer 2 raises this point, and following their suggestion, we have now added several sentences to the results section where we describe the age-related patterns in words. For quadratic relationships, the patterns are more difficult to put concisely into words as the increase and/or decrease in social network measures accelerates or decelerates when you move further or closer from the maximum or minimum. In these cases, we discussed in which age categories the maxima and minima of the social network measures fell. For significant linear effects, we discuss the changes with age in terms of percentages, as suggested by the reviewer. For example: 

L332-335: “In males, out-strength declined linearly with age (Out-strength: β = -0.002 ± 0.001 SE, t(128.3) = -2.911, p = 0.004), indicating that for each 10-year increase in age, males decrease their time spent on grooming others with 2%. To put this in perspective, the average out-strength for males across all ages is 0.079, indicating that about 8% of their time is spent on grooming others.”

A related point is that the authors were unable to report confidence intervals for their models. I strongly feel reporting intervals helps to biologically ground the results, so that readers appreciate the scientific implications of the findings. Confidence intervals can usually be extracted from most popular software packages. I understand lmerTest runs tests on lme4 models, and lme4 supports the computation of confidence intervals. I would encourage the authors to revisit this point and see if intervals could be reported. I feel it would really help readers to appreciate these interesting findings.

This is a good point from the reviewer, as by an error on our side we forgot to report standard errors or confidence intervals in our results section and only included standard errors in the supplement. We have now added the standard errors of the estimates in our results section and we have added confidence intervals in the tables in the supplementary information, which contain all of the test statistics per model.

I’m looking forward to seeing the paper published after these changes are made.

We again would like to thank the reviewer for their constructive feedback on the current and previous version of the manuscript.

---

## [Decision Letter · Decision Letter 2]

29 Mar 2023

Multi-group analysis of grooming network position in a highly social primate

PONE-D-22-29445R2

Dear Dr. Torfs,

We’re pleased to inform you that your manuscript has been judged scientifically suitable for publication and will be formally accepted for publication once it meets all outstanding technical requirements.

Kind regards,

Daniel Redhead

Academic Editor

PLOS ONE

Additional Editor Comments (optional):

The authors have not addressed all issues raised by the editor and reviewers. The manuscript is now in an acceptable condition for publication, and while it does have some limitations, those limitations are outlined clearly and transparently in the discussion. 

Reviewers' comments:

Reviewer's Responses to Questions

**Comments to the Author**

1. If the authors have adequately addressed your comments raised in a previous round of review and you feel that this manuscript is now acceptable for publication, you may indicate that here to bypass the “Comments to the Author” section, enter your conflict of interest statement in the “Confidential to Editor” section, and submit your "Accept" recommendation.

Reviewer #1: All comments have been addressed

2. Is the manuscript technically sound, and do the data support the conclusions?

Reviewer #1: Yes

3. Has the statistical analysis been performed appropriately and rigorously? 

Reviewer #1: Yes

4. Have the authors made all data underlying the findings in their manuscript fully available?

Reviewer #1: Yes

5. Is the manuscript presented in an intelligible fashion and written in standard English?

Reviewer #1: Yes

6. Review Comments to the Author

Reviewer #1: The authors have once again done a good job of addressing the comments in my previous review.

A few very minor points here from reading through the revised version:

L236-237: Correlations only for groups of 5 or larger? This is not clear as that decision is only highlighted in the subsequent section

L305-315: This could probably be shortened a little/made more clear – it seems the key points are that the authors only retained linear hierarchies but used an alpha level of 0.1 due to limitations in their data (dyads with few/no interactions).

Results: I think the convention is to focus on estimates and confidence intervals (whether they overlap zero or not) after AIC model selection rather than necessarily using p values. That being said, I am happy to leave this at the discretion of the authors, as I don’t think it is important here.

L340-341: It is probably easier to say no quadratic relationship here given the clearly null result?

L486: I think there is a typo in this sentence

L496: Need to be clear this has been shown, but not necessarily in bonobos (e.g. the study cited here is for macaques)

All of these are only small, cosmetic changes.

7. PLOS authors have the option to publish the peer review history of their article (what does this mean?). If published, this will include your full peer review and any attached files.

Reviewer #1: No

---

## [Editor Report · Acceptance letter]

3 Apr 2023

PONE-D-22-29445R2 

Multi-group analysis of grooming network position in a highly social primate 

Dear Dr. Torfs:

I'm pleased to inform you that your manuscript has been deemed suitable for publication in PLOS ONE. Congratulations! Your manuscript is now with our production department. 

Kind regards, 

on behalf of

Dr. Daniel Redhead 

Academic Editor

PLOS ONE